# Knowledge Graph Completion by Intermediate Variables Regularization

**Changyi Xiao, Yixin Cao**[*]
School of Computer Science, Fudan University
changyi_xiao@fudan.edu.cn, caoyixin2011@gmail.com

## Abstract

Knowledge graph completion (KGC) can be framed as a 3-order binary tensor completion task. Tensor decomposition-based (TDB) models have demonstrated strong performance in KGC. In this paper, we provide a summary of existing TDB models and derive a general form for them, serving as a foundation for further exploration of TDB models. Despite the expressiveness of TDB models, they are prone to overfitting. Existing regularization methods merely minimize the norms of embeddings to regularize the model, leading to suboptimal performance. Therefore, we propose a novel regularization method for TDB models that addresses this limitation. The regularization is applicable to most TDB models and ensures tractable computation. Our method minimizes the norms of intermediate variables involved in the different ways of computing the predicted tensor. To support our regularization method, we provide a theoretical analysis that proves its effect in promoting low trace norm of the predicted tensor to reduce overfitting. Finally, we conduct experiments to verify the effectiveness of our regularization technique as well as the reliability of our theoretical analysis. The code is available at https://github.com/changyi7231/IVR.

## 1 Introduction

A knowledge graph (KG) can be represented as a 3rd-order binary tensor, in which each entry corresponds to a triplet of the form $(head\ entity, relation, tail\ entity)$. A value of 1 denotes a known true triplet, while 0 denotes a false triplet. Despite containing a large number of known triplets, KGs are often incomplete, with many triplets missing. Consequently, the 3rd-order tensors representing the KGs are incomplete. The objective of knowledge graph completion (KGC) is to infer the true or false values of the missing triplets based on the known ones, i.e., to predict which of the missing entries in the tensor are 1 or 0.

A number of models have been proposed for KGC, which can be classified into translation-based models, tensor decomposition-based (TDB) models and neural networks models [Zhang et al., 2021]. We only focus on TDB models in this paper due to their wide applicability and great performance [Lacroix et al., 2018, Zhang et al., 2020]. TDB models can be broadly categorized into two groups: CANDECOMP/PARAFAC (CP) decomposition-based models, including CP [Lacroix et al., 2018], DistMult [Yang et al., 2014] and ComplEx [Trouillon et al., 2017], and Tucker decomposition-based models, including SimplE [Kazemi and Poole, 2018], ANALOGY [Liu et al., 2017], QuatE [Zhang et al., 2019] and TuckER [Balažević et al., 2019]. To provide a thorough understanding of TDB models, we present a summary of existing models and derive a general form that unifies them, which provides a fundamental basis for further exploration of TDB models.

---

[*]Corresponding author.

38th Conference on Neural Information Processing Systems (NeurIPS 2024).

TDB models have been proven to be theoretically fully expressive [Trouillon et al., 2017, Kazemi and Poole, 2018, Balaževič et al., 2019], implying they can represent any real-valued tensor. However, in practice, TDB models frequently fall prey to severe overfitting. To counteract this issue, various regularization techniques have been employed in KGC. One commonly used technique is the squared Frobenius norm regularization [Nickel et al., 2011, Yang et al., 2014, Trouillon et al., 2017]. Lacroix et al. [2018] proposed another regularization, N3 norm regularization, based on the tensor nuclear $p$-norm, which outperforms the squared Frobenius norm in terms of performance. Additionally, Zhang et al. [2020] introduced DURA, a regularization technique based on the duality of TDB models and distance-based models that results in significant improvements on benchmark datasets. Nevertheless, N3 and DURA rely on CP decomposition, limiting their applicability to CP [Lacroix et al., 2018] and ComplEx [Trouillon et al., 2017]. As a result, there is a pressing need for a regularization technique that is widely applicable and can effectively alleviate the overfitting issue.

In this paper, we introduce a novel regularization method for KGC to improve the performance of TDB models. Our regularization focuses on preventing overfitting while maintaining the expressiveness of TDB models as much as possible. It is applicable to most TDB models, while also ensuring tractable computation. Existing regularization methods for KGC rely on minimizing the norms of embeddings to regularize the model [Yang et al., 2014, Lacroix et al., 2018], leading to suboptimal performance. To achieve superior performance, we present an intermediate variables regularization (IVR) approach that minimizes the norms of intermediate variables involved in the processes of computing the predicted tensor of TDB models. Additionally, our approach fully considers the computing ways because different ways of computing the predicted tensor may generate different intermediate variables.

To support the efficacy of our regularization approach, we further provide a theoretical analysis. We prove that our regularization is an upper bound of the overlapped trace norm [Tomioka et al., 2011]. The overlapped trace norm is the sum of the trace norms of the unfolding matrices along each mode of a tensor [Kolda and Bader, 2009], which can be considered as a surrogate measure of the rank of a tensor. Thus, the overlapped trace norm reflects the correlation among entities and relations, which can pose a constraint to jointly entities and relations embeddings learning. In specific, entities and relations in KGs are usually highly correlated. For example, some relations are mutual inverse relations, or a relation may be a composition of another two relations [Zhang et al., 2021]. Through minimizing the upper bound of the overlapped trace norm, we encourage a high correlation among entities and relations, which brings strong regularization and alleviates the overfitting problem.

The main contributions of this paper are listed below:

1. We present a detailed overview of a wide range of TDB models and establish a general form to serve as a foundation for further TDB model analysis.

2. We introduce a new regularization approach for TDB models based on the general form to mitigate the overfitting issue, which is notable for its generality and effectiveness.

3. We provide a theoretical proof of the efficacy of our regularization and validate its practical utility through experiments.

## 2   Related Work

**Tensor Decomposition Based Models**   Research in KGC has been vast, with TDB models garnering attention due to their superior performance. The two primary TDB approaches that have been extensively studied are CP decomposition [Hitchcock, 1927] and Tucker decomposition [Tucker, 1966]. CP decomposition represents a tensor as a sum of $n$ rank-one tensors, while Tucker decomposition decomposes a tensor into a core tensor and a set of matrices. Kolda and Bader [2009] shows more details about CP decomposition and Tucker decomposition.

Several techniques have been developed for applying CP decomposition and Tucker decomposition in KGC. For instance, Lacroix et al. [2018] employed the original CP decomposition, whereas DistMult [Yang et al., 2014], a variant of CP decomposition, made the embedding matrices of head entities and tail entities identical to simplify the model. SimplE [Kazemi and Poole, 2018] tackled the problem of independence among the embeddings of head and tail entities within CP decomposition. ComplEx [Trouillon et al., 2017] extended DistMult to the complex space to handle asymmetric relations. QuatE [Zhang et al., 2019] explored hypercomplex space to further enhance KGC. Other techniques

include HolE [Nickel et al., 2016] proposed by Nickel et al. [2016], which operates on circular correlation, and ANALOGY [Liu et al., 2017], which explicitly utilizes the analogical structures of KGs. Notably, Liu et al. [2017] confirmed that HolE is equivalent to ComplEx. Additionally, Balažević et al. [2019] introduced TuckER, which is based on the Tucker decomposition and has achieved state-of-the-art performance across various benchmark datasets for KGC. Nickel et al. [2011] proposed a three-way decomposition RESCAL over each relational slice of the tensor. You can refer to [Zhang et al., 2021] or [Ji et al., 2021] for more detailed discussion about KGC models or TDB models.

**Regularization** Although TDB models are highly expressive [Trouillon et al., 2017, Kazemi and Poole, 2018, Balažević et al., 2019], they can suffer severely from overfitting in practice. Consequently, several regularization approaches have been proposed. A common regularization approach is to apply the squared Frobenius norm to the model parameters [Nickel et al., 2011, Yang et al., 2014, Trouillon et al., 2017]. However, this approach does not correspond to a proper tensor norm, as shown by Lacroix et al. [2018]. Therefore, they proposed a novel regularization method, N3, based on the tensor nuclear 3-norm, which is an upper bound of the tensor nuclear norm. Likewise, Zhang et al. [2020] introduced DURA, a regularization method that exploits the duality of TDB models and distance-based models, and serves as an upper bound of the tensor nuclear 2-norm. However, both N3 and DURA are derived from the CP decomposition, and thus are only applicable to CP and ComplEx models.

## 3 Methods

In Section 3.1, we begin by providing an overview of existing TDB models and derive a general form for them. Thereafter, we present our intermediate variables regularization (IVR) approach in Section 3.2. Finally, in Section 3.3, we provide theoretical analysis to support the efficacy of our proposed regularization technique.

### 3.1 General Form

To facilitate the subsequent theoretical analysis, we initially provide a summary of existing TDB models and derive a general form for them. Given a set of entities $\mathcal{E}$ and a set of relations $\mathcal{R}$, a KG contains a set of triplets $\mathcal{S} = \{(i, j, k)\} \subset \mathcal{E} \times \mathcal{R} \times \mathcal{E}$. Let $\boldsymbol{X} \in \{0, 1\}^{|\mathcal{E}| \times |\mathcal{R}| \times |\mathcal{E}|}$ represent the KG tensor, with $\boldsymbol{X}_{ijk} = 1$ iff $(i, j, k) \in \mathcal{S}$, where $|\mathcal{E}|$ and $|\mathcal{R}|$ denote the number of entities and relations, respectively. Let $\boldsymbol{H} \in \mathbb{R}^{|\mathcal{E}| \times D}$, $\boldsymbol{R} \in \mathbb{R}^{|\mathcal{R}| \times D}$ and $\boldsymbol{T} \in \mathbb{R}^{|\mathcal{E}| \times D}$ be the embedding matrices of head entities, relations and tail entities, respectively, where $D$ is the embedding dimension.

Various TDB models can be attained by partitioning the embedding matrices into $P$ parts. We reshape $\boldsymbol{H} \in \mathbb{R}^{|\mathcal{E}| \times D}$, $\boldsymbol{R} \in \mathbb{R}^{|\mathcal{R}| \times D}$ and $\boldsymbol{T} \in \mathbb{R}^{|\mathcal{E}| \times D}$ into $\boldsymbol{H} \in \mathbb{R}^{|\mathcal{E}| \times (D/P) \times P}$, $\boldsymbol{R} \in \mathbb{R}^{|\mathcal{R}| \times (D/P) \times P}$ and $\boldsymbol{T} \in \mathbb{R}^{|\mathcal{E}| \times (D/P) \times P}$, respectively, where $P$ is the number of parts we partition. For different $P$, we can get different TDB models.

**CP/DistMult** Let $P = 1$, CP [Lacroix et al., 2018] can be represented as

$$\boldsymbol{X}_{ijk} = \langle \boldsymbol{H}_{i:1}, \boldsymbol{R}_{j:1}, \boldsymbol{T}_{k:1} \rangle := \sum_{d=1}^{D/P} \boldsymbol{H}_{id1} \boldsymbol{R}_{jd1} \boldsymbol{T}_{kd1}$$

where $\langle \cdot, \cdot, \cdot \rangle$ is the dot product of three vectors. DistMult [Yang et al., 2014], a particular case of CP, which shares the embedding matrices of head entities and tail entities, i.e., $\boldsymbol{H} = \boldsymbol{T}$.

**ComplEx/HolE** Let $P = 2$, ComplEx [Trouillon et al., 2017] can be represented as

$$\boldsymbol{X}_{ijk} = \langle \boldsymbol{H}_{i:1}, \boldsymbol{R}_{j:1}, \boldsymbol{T}_{k:1} \rangle + \langle \boldsymbol{H}_{i:2}, \boldsymbol{R}_{j:1}, \boldsymbol{T}_{k:2} \rangle + \langle \boldsymbol{H}_{i:1}, \boldsymbol{R}_{j:2}, \boldsymbol{T}_{k:2} \rangle - \langle \boldsymbol{H}_{i:2}, \boldsymbol{R}_{j:2}, \boldsymbol{T}_{k:1} \rangle$$

Liu et al. [2017] proved that HolE [Nickel et al., 2011] is equivalent to ComplEx.

**SimplE** Let $P = 2$, SimplE [Kazemi and Poole, 2018] can be represented as

$$\boldsymbol{X}_{ijk} = \langle \boldsymbol{H}_{i:1}, \boldsymbol{R}_{j:1}, \boldsymbol{T}_{k:2} \rangle + \langle \boldsymbol{H}_{i:2}, \boldsymbol{R}_{j:2}, \boldsymbol{T}_{k:1} \rangle$$

**ANALOGY** Let $P = 4$, ANALOGY [Liu et al., 2017] can be represented as

$$\boldsymbol{X}_{ijk} = \langle \boldsymbol{H}_{i:1}, \boldsymbol{R}_{j:1}, \boldsymbol{T}_{k:1} \rangle + \langle \boldsymbol{H}_{i:2}, \boldsymbol{R}_{j:2}, \boldsymbol{T}_{k:2} \rangle + \langle \boldsymbol{H}_{i:3}, \boldsymbol{R}_{j:3}, \boldsymbol{T}_{k:3} \rangle + \langle \boldsymbol{H}_{i:3}, \boldsymbol{R}_{j:4}, \boldsymbol{T}_{k:4} \rangle$$
$$+ \langle \boldsymbol{H}_{i:4}, \boldsymbol{R}_{j:3}, \boldsymbol{T}_{k:4} \rangle - \langle \boldsymbol{H}_{i:4}, \boldsymbol{R}_{j:4}, \boldsymbol{T}_{k:3} \rangle$$

**QuatE** Let $P = 4$, QuatE [Zhang et al., 2019] can be represented as

$$\boldsymbol{X}_{ijk} = \langle \boldsymbol{H}_{i:1}, \boldsymbol{R}_{j:1}, \boldsymbol{T}_{k:1} \rangle - \langle \boldsymbol{H}_{i:2}, \boldsymbol{R}_{j:2}, \boldsymbol{T}_{k:1} \rangle - \langle \boldsymbol{H}_{i:3}, \boldsymbol{R}_{j:3}, \boldsymbol{T}_{k:1} \rangle - \langle \boldsymbol{H}_{i:4}, \boldsymbol{R}_{j:4}, \boldsymbol{T}_{k:1} \rangle$$
$$+ \langle \boldsymbol{H}_{i:1}, \boldsymbol{R}_{j:2}, \boldsymbol{T}_{k:2} \rangle + \langle \boldsymbol{H}_{i:2}, \boldsymbol{R}_{j:1}, \boldsymbol{T}_{k:2} \rangle + \langle \boldsymbol{H}_{i:3}, \boldsymbol{R}_{j:4}, \boldsymbol{T}_{k:2} \rangle - \langle \boldsymbol{H}_{i:4}, \boldsymbol{R}_{j:3}, \boldsymbol{T}_{k:2} \rangle$$
$$+ \langle \boldsymbol{H}_{i:1}, \boldsymbol{R}_{j:3}, \boldsymbol{T}_{k:3} \rangle - \langle \boldsymbol{H}_{i:2}, \boldsymbol{R}_{j:4}, \boldsymbol{T}_{k:3} \rangle + \langle \boldsymbol{H}_{i:3}, \boldsymbol{R}_{j:1}, \boldsymbol{T}_{k:3} \rangle + \langle \boldsymbol{H}_{i:4}, \boldsymbol{R}_{j:2}, \boldsymbol{T}_{k:3} \rangle$$
$$+ \langle \boldsymbol{H}_{i:1}, \boldsymbol{R}_{j:4}, \boldsymbol{T}_{k:4} \rangle + \langle \boldsymbol{H}_{i:2}, \boldsymbol{R}_{j:3}, \boldsymbol{T}_{k:4} \rangle - \langle \boldsymbol{H}_{i:3}, \boldsymbol{R}_{j:2}, \boldsymbol{T}_{k:4} \rangle + \langle \boldsymbol{H}_{i:4}, \boldsymbol{R}_{j:1}, \boldsymbol{T}_{k:4} \rangle$$

**TuckER** Let $P = D$, TuckER [Balažević et al., 2019] can be represented as

$$\boldsymbol{X}_{ijk} = \sum_{l=1}^{P} \sum_{m=1}^{P} \sum_{n=1}^{P} \boldsymbol{W}_{lmn} \boldsymbol{H}_{i1l} \boldsymbol{R}_{j1m} \boldsymbol{T}_{k1n}$$

where $\boldsymbol{W} \in \mathbb{R}^{P \times P \times P}$ is the core tensor.

**General Form** Through our analysis, we observe that all TDB models can be expressed as a linear combination of several dot product. The key distinguishing factors among these models are the choice of the number of parts $P$ and the core tensor $\boldsymbol{W}$. The number of parts $P$ determines the dimensions of the dot products of the embeddings, while the core tensor $\boldsymbol{W}$ determines the strength of the dot products. It is important to note that TuckER uses a parameter tensor as its core tensor, whereas the core tensors of other models are predetermined constant tensors. Therefore, we can derive a general form of these models as

$$\boldsymbol{X}_{ijk} = \sum_{l=1}^{P} \sum_{m=1}^{P} \sum_{n=1}^{P} \boldsymbol{W}_{lmn} \langle \boldsymbol{H}_{i:l}, \boldsymbol{R}_{j:m}, \boldsymbol{T}_{k:n} \rangle = \sum_{l=1}^{P} \sum_{m=1}^{P} \sum_{n=1}^{P} \boldsymbol{W}_{lmn} (\sum_{d=1}^{D/P} \boldsymbol{H}_{idl} \boldsymbol{R}_{jdm} \boldsymbol{T}_{kdn})$$
$$= \sum_{d=1}^{D/P} (\sum_{l=1}^{P} \sum_{m=1}^{P} \sum_{n=1}^{P} \boldsymbol{W}_{lmn} \boldsymbol{H}_{idl} \boldsymbol{R}_{jdm} \boldsymbol{T}_{kdn}) \tag{1}$$

or

$$\boldsymbol{X} = \sum_{d=1}^{D/P} \boldsymbol{W} \times_1 \boldsymbol{H}_{:d:} \times_2 \boldsymbol{R}_{:d:} \times_3 \boldsymbol{T}_{:d:} \tag{2}$$

where $\times_n$ is the mode-$n$ product [Kolda and Bader, 2009], and $\boldsymbol{W} \in \mathbb{R}^{P \times P \times P}$ is the core tensor, which can be a parameter tensor or a predetermined constant tensor. The general form Eq.(2) can also be considered as a sum of $D/P$ TuckER decompositions, which is also called block-term decomposition [De Lathauwer, 2008]. Eq.(2) is a block-term decomposition with a shared core tensor $\boldsymbol{W}$. This general form is easy to understand, facilitates better understanding of TDB models and paves the way for further exploration of TDB models. The general form presents a unified view of TDB models and helps the researchers understand the relationship between different TDB models. Moreover, the general form motivates the researchers to propose new methods and establish unified theoretical frameworks that are applicable to most TDB models. Our proposed regularization in Section 3.2 and the theoretical analysis Section 3.3 are examples of such contributions.

**The Number of Parameters and Computational Complexity** The parameters of Eq.(2) come from two parts, the core tensor $\boldsymbol{W}$ and the embedding matrices $\boldsymbol{H}$, $\boldsymbol{R}$ and $\boldsymbol{T}$. The number of parameters of the core tensor $\boldsymbol{W}$ is equal to $P^3$ if $\boldsymbol{W}$ is a parameter tensor and otherwise equal to 0. The number of parameters of the embedding matrices is equal to $|\mathcal{E}|D + |\mathcal{R}|D$ if $\boldsymbol{H} = \boldsymbol{T}$ and otherwise equal to $2|\mathcal{E}|D + |\mathcal{R}|D$. The computational complexity of Eq.(2) is equal to $\mathcal{O}(DP^2|\mathcal{E}|^2|\mathcal{R}|)$. The larger the number of parts $P$, the more expressive the model and the more the computation. Therefore, the choice of $P$ is a trade-off between expressiveness and computation.

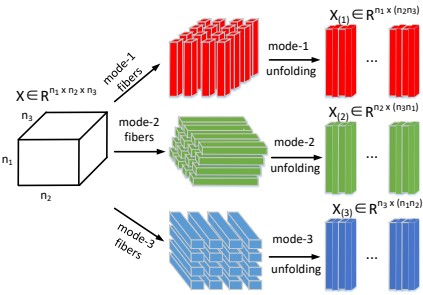

Figure 1: Left shows a 3rd order tensor. Middle describes the corresponding mode-$i$ fibers of the tensor. Fibers are the higher-order analogue of matrix rows and columns. A fiber is defined by fixing every index but one. Right describes the corresponding mode-$i$ unfolding of the tensor. The mode-$i$ unfolding of a tensor arranges the mode-$i$ fibers to be the columns of the resulting matrix.

**TuckER and Eq.(2)**   TuckER [Balažević et al., 2019] also demonstrated that TDB models can be represented as a Tucker decomposition by setting specific core tensors $\boldsymbol{W}$. Nevertheless, we must stress that TuckER does not explicitly consider the number of parts $P$ and the core tensor $\boldsymbol{W}$, which are pertinent to the number of parameters and computational complexity of TDB models. Moreover, in Appendix A, we demonstrate that the conditions for a TDB model to learn logical rules are also dependent on $P$ and $\boldsymbol{W}$. By selecting appropriate $P$ and $\boldsymbol{W}$, TDB models can be able to learn symmetry rules, antisymmetry rules, and inverse rules.

### 3.2   Intermediate Variables Regularization

TDB models are theoretically fully expressive [Trouillon et al., 2017, Kazemi and Poole, 2018, Balažević et al., 2019], which can represent any real-valued tensor. However, TDB models suffer from the overfitting problem in practice [Lacroix et al., 2018]. Therefore, several regularization methods have been proposed, such as squared Frobenius norm method [Yang et al., 2014] and nuclear 3-norm method [Lacroix et al., 2018], which minimize the norms of the embeddings $\{\boldsymbol{H}, \boldsymbol{R}, \boldsymbol{T}\}$ to regularize the model. Nonetheless, merely minimizing the embeddings tends to have suboptimal impacts on the model performance. To enhance the model performance, we introduce a new regularization method that minimizes the norms of the intermediate variables involved in the processes of computing $\boldsymbol{X}$. To ensure the broad applicability of our method, our regularization is rooted in the general form of TDB models Eq.(2).

To compute $\boldsymbol{X}$ in Eq.(2), we can first compute the intermediate variable $\boldsymbol{W} \times_1 \boldsymbol{H}_{:d:} \times_2 \boldsymbol{R}_{:d:}$, and then combine $\boldsymbol{T}_{:d:}$ to compute $\boldsymbol{X}$. Thus, in addition to minimizing the norm of $\boldsymbol{T}_{:d:}$, we also need to minimize the norm of $\boldsymbol{W} \times_1 \boldsymbol{H}_{:d:} \times_2 \boldsymbol{R}_{:d:}$. Since different ways of computing $\boldsymbol{X}$ can result in different intermediate variables, we fully consider the computing ways of $\boldsymbol{X}$. Eq.(2) can also be written as:

$$\boldsymbol{X} = \sum_{d=1}^{D/P} (\boldsymbol{W} \times_2 \boldsymbol{R}_{:d:} \times_3 \boldsymbol{T}_{:d:}) \times_1 \boldsymbol{H}_{:d:},$$

$$\boldsymbol{X} = \sum_{d=1}^{D/P} (\boldsymbol{W} \times_3 \boldsymbol{T}_{:d:} \times_1 \boldsymbol{H}_{:d:}) \times_2 \boldsymbol{R}_{:d:}$$

Thus, we also need to minimize the norms of intermediate variables $\{\boldsymbol{W} \times_2 \boldsymbol{R}_{:d:} \times_3 \boldsymbol{T}_{:d:}, \boldsymbol{W} \times_3 \boldsymbol{T}_{:d:} \times_1 \boldsymbol{H}_{:d:}\}$ and $\{\boldsymbol{H}_{:d:}, \boldsymbol{R}_{:d:}\}$. In summary, we should minimize the (power of Frobenius) norms $\{\|\boldsymbol{H}_{:d:}\|_F^\alpha, \|\boldsymbol{R}_{:d:}\|_F^\alpha, \|\boldsymbol{T}_{:d:}\|_F^\alpha\}$ and $\{\|\boldsymbol{W} \times_1 \boldsymbol{H}_{:d:} \times_2 \boldsymbol{R}_{:d:}\|_F^\alpha, \|\boldsymbol{W} \times_2 \boldsymbol{R}_{:d:} \times_3 \boldsymbol{T}_{:d:}\|_F^\alpha, \|\boldsymbol{W} \times_3 \boldsymbol{T}_{:d:} \times_1 \boldsymbol{H}_{:d:}\|_F^\alpha\}$, where $\alpha$ is the power of the norms.

Since computing $\boldsymbol{X}$ is equivalent to computing $\boldsymbol{X}_{(1)}$ or $\boldsymbol{X}_{(2)}$ or $\boldsymbol{X}_{(3)}$, we can also minimize the norms of intermediate variables involved in the processes of computing $\boldsymbol{X}_{(1)}$, $\boldsymbol{X}_{(2)}$ and $\boldsymbol{X}_{(3)}$, where $\boldsymbol{X}_{(n)}$ is the mode-$n$ unfolding of a tensor $\boldsymbol{X}$ [Kolda and Bader, 2009]. See Figure 1 for an example

of the notation $\boldsymbol{X}_{(n)}$. We can represent $\boldsymbol{X}_{(1)}$, $\boldsymbol{X}_{(2)}$ and $\boldsymbol{X}_{(3)}$ as [Kolda and Bader, 2009]:

$$\boldsymbol{X}_{(1)} = \sum_{d=1}^{D/P} (\boldsymbol{W} \times_1 \boldsymbol{H}_{:d:})_{(1)} (\boldsymbol{T}_{:d:} \otimes \boldsymbol{R}_{:d:})^T,$$

$$\boldsymbol{X}_{(2)} = \sum_{d=1}^{D/P} (\boldsymbol{W} \times_2 \boldsymbol{R}_{:d:})_{(2)} (\boldsymbol{T}_{:d:} \otimes \boldsymbol{H}_{:d:})^T,$$

$$\boldsymbol{X}_{(3)} = \sum_{d=1}^{D/P} (\boldsymbol{W} \times_3 \boldsymbol{T}_{:d:})_{(3)} (\boldsymbol{R}_{:d:} \otimes \boldsymbol{H}_{:d:})^T$$

where $\otimes$ is the Kronecker product. Thus, the intermediate variables include $\{\boldsymbol{W} \times_1 \boldsymbol{H}_{:d:}, \boldsymbol{W} \times_2 \boldsymbol{R}_{:d:}, \boldsymbol{W} \times_3 \boldsymbol{T}_{:d:}\}$ and $\{\boldsymbol{T}_{:d:} \otimes \boldsymbol{R}_{:d:}, \boldsymbol{T}_{:d:} \otimes \boldsymbol{H}_{:d:}, \boldsymbol{R}_{:d:} \otimes \boldsymbol{H}_{:d:}\}$. Therefore, we should minimize the (power of Frobenius) norms $\{\|\boldsymbol{W} \times_1 \boldsymbol{H}_{:d:}\|_F^\alpha, \|\boldsymbol{W} \times_2 \boldsymbol{R}_{:d:}\|_F^\alpha, \|\boldsymbol{W} \times_3 \boldsymbol{T}_{:d:}\|_F^\alpha\}$ and $\{\|\boldsymbol{T}_{:d:} \otimes \boldsymbol{R}_{:d:}\|_F^\alpha = \|\boldsymbol{T}_{:d:}\|_F^\alpha \|\boldsymbol{R}_{:d:}\|_F^\alpha, \|\boldsymbol{T}_{:d:} \otimes \boldsymbol{H}_{:d:}\|_F^\alpha = \|\boldsymbol{T}_{:d:}\|_F^\alpha \|\boldsymbol{H}_{:d:}\|_F^\alpha, \|\boldsymbol{R}_{:d:} \otimes \boldsymbol{H}_{:d:}\|_F^\alpha = \|\boldsymbol{R}_{:d:}\|_F^\alpha \|\boldsymbol{H}_{:d:}\|_F^\alpha\}$.

Our Intermediate Variables Regularization (IVR) is defined as a combination of all these norms:

$$\begin{aligned} \text{reg}(\boldsymbol{X}) = & \sum_{d=1}^{D/P} \lambda_1 (\|\boldsymbol{H}_{:d:}\|_F^\alpha + \|\boldsymbol{R}_{:d:}\|_F^\alpha + \|\boldsymbol{T}_{:d:}\|_F^\alpha) \\ & + \lambda_2 (\|\boldsymbol{T}_{:d:}\|_F^\alpha \|\boldsymbol{R}_{:d:}\|_F^\alpha + \|\boldsymbol{T}_{:d:}\|_F^\alpha \|\boldsymbol{H}_{:d:}\|_F^\alpha + \|\boldsymbol{R}_{:d:}\|_F^\alpha \|\boldsymbol{H}_{:d:}\|_F^\alpha) \\ & + \lambda_3 (\|\boldsymbol{W} \times_1 \boldsymbol{H}_{:d:}\|_F^\alpha + \|\boldsymbol{W} \times_2 \boldsymbol{R}_{:d:}\|_F^\alpha + \|\boldsymbol{W} \times_3 \boldsymbol{T}_{:d:}\|_F^\alpha) \\ & + \lambda_4 (\|\boldsymbol{W} \times_2 \boldsymbol{R}_{:d:} \times_3 \boldsymbol{T}_{:d:}\|_F^\alpha + \|\boldsymbol{W} \times_3 \boldsymbol{T}_{:d:} \times_1 \boldsymbol{H}_{:d:}\|_F^\alpha + \|\boldsymbol{W} \times_1 \boldsymbol{H}_{:d:} \times_2 \boldsymbol{R}_{:d:}\|_F^\alpha) \end{aligned} \quad (3)$$

where $\{\lambda_i > 0 | i = 1, 2, 3, 4\}$ are the regularization coefficients.

In conclusion, our proposed regularization term is the sum of the norms of variables involved in the different ways of computing the tensor $\boldsymbol{X}$.

We can easily get the weighted version of Eq.(3), in which the regularization term corresponding to the sampled training triplets only [Lacroix et al., 2018, Zhang et al., 2020]. For a training triplet $(i, j, k)$, the weighted version of Eq.(3) is as follows:

$$\begin{aligned} \text{reg}(\boldsymbol{X}_{ijk}) = & \sum_{d=1}^{D/P} \lambda_1 (\|\boldsymbol{H}_{id:}\|_F^\alpha + \|\boldsymbol{R}_{jd:}\|_F^\alpha + \|\boldsymbol{T}_{kd:}\|_F^\alpha) \\ & + \lambda_2 (\|\boldsymbol{T}_{kd:}\|_F^\alpha \|\boldsymbol{R}_{jd:}\|_F^\alpha + \|\boldsymbol{T}_{kd:}\|_F^\alpha \|\boldsymbol{H}_{id:}\|_F^\alpha + \|\boldsymbol{R}_{jd:}\|_F^\alpha \|\boldsymbol{H}_{id:}\|_F^\alpha) \\ & + \lambda_3 (\|\boldsymbol{W} \times_1 \boldsymbol{H}_{id:}\|_F^\alpha + \|\boldsymbol{W} \times_2 \boldsymbol{R}_{jd:}\|_F^\alpha + \|\boldsymbol{W} \times_3 \boldsymbol{T}_{kd:}\|_F^\alpha) \\ & + \lambda_4 (\|\boldsymbol{W} \times_2 \boldsymbol{R}_{jd:} \times_3 \boldsymbol{T}_{kd:}\|_F^\alpha + \|\boldsymbol{W} \times_3 \boldsymbol{T}_{kd:} \times_1 \boldsymbol{H}_{id:}\|_F^\alpha + \|\boldsymbol{W} \times_1 \boldsymbol{H}_{id:} \times_2 \boldsymbol{R}_{jd:}\|_F^\alpha) \end{aligned} \quad (4)$$

The computational complexity of Eq.(4) is the same as that of Eq.(1), i.e., $\mathcal{O}(DP^2)$, which ensures that our regularization is computationally tractable.

The hyper-parameters $\lambda_i$ make IVR scalable. We can easily reduce the number of hyper-parameters by setting some of them zero or equal. The hyper-parameters make us able to achieve a balance between performance and efficiency as shown in Section 4.3. We set $\lambda_1 = \lambda_3$ and $\lambda_2 = \lambda_4$ for all models to reduce the number of hyper-parameters. You can refer to Appendix C for more details about the setting of hyper-parameters.

We use the same loss function, multiclass log-loss function, as in [Lacroix et al., 2018]. For a training triplet $(i, j, k)$, our loss function is

$$\ell(\boldsymbol{X}_{ijk}) = -\boldsymbol{X}_{ijk} + \log(\sum_{k'=1}^{|\mathcal{E}|} \exp(\boldsymbol{X}_{ijk'})) + \text{reg}(\boldsymbol{X}_{ijk})$$

At test time, we use $\boldsymbol{X}_{i,j,:}$ to rank tail entities for a query $(i, j, ?)$.

## 3.3 Theoretical Analysis

To support the effectiveness of our regularization IVR, we provide a deeper theoretical analysis of its properties. The establishment of the theoretical framework of IVR is inspired by Lemma 1 in

Appendix B, which relates the Frobenius norm and the trace norm of a matrix. Lemma 1 shows that the trace norm of a matrix is an upper bound of a function of several Frobenius norms of intermediate variables, which prompts us to establish the relationship between IVR and trace norm. Based on Lemma 1, we prove that IVR serves as an upper bound for the overlapped trace norm of the predicted tensor, which promotes the low nuclear norm of the predicted tensor to regularize the model.

The overlapped trace norm [Kolda and Bader, 2009] for a 3rd-order tensor is defined as:

$$L(\boldsymbol{X};\alpha) := \|\boldsymbol{X}_{(1)}\|_*^{\alpha/2} + \|\boldsymbol{X}_{(2)}\|_*^{\alpha/2} + \|\boldsymbol{X}_{(3)}\|_*^{\alpha/2}$$

where $\alpha$ is the power coefficient in Eq.(3). $\|\boldsymbol{X}_{(1)}\|_*, \|\boldsymbol{X}_{(2)}\|_*$ and $\|\boldsymbol{X}_{(3)}\|_*$ are the matrix trace norms of $\boldsymbol{X}_{(1)}, \boldsymbol{X}_{(2)}$ and $\boldsymbol{X}_{(3)}$, respectively, which are the sums of singular values of the respective matrices. The matrix trace norm is widely used as a convex surrogate for matrix rank due to the non-differentiability of matrix rank [Goldfarb and Qin, 2014, Lu et al., 2016, Mu et al., 2014]. Thus, $L(\boldsymbol{X};\alpha)$ serves as a surrogate for $\mathrm{rank}(\boldsymbol{X}_{(1)})^{\alpha/2} + \mathrm{rank}(\boldsymbol{X}_{(2)})^{\alpha/2} + \mathrm{rank}(\boldsymbol{X}_{(3)})^{\alpha/2}$, where $\mathrm{rank}(\boldsymbol{X}_{(1)}), \mathrm{rank}(\boldsymbol{X}_{(2)})$ and $\mathrm{rank}(\boldsymbol{X}_{(3)})$ are the matrix ranks of $\boldsymbol{X}_{(1)}, \boldsymbol{X}_{(2)}$ and $\boldsymbol{X}_{(3)}$, respectively. In KGs, each head entity, each relation and each tail entity uniquely corresponds to a row of $\boldsymbol{X}_{(1)}, \boldsymbol{X}_{(2)}$ and $\boldsymbol{X}_{(3)}$, respectively. Therefore, $\mathrm{rank}(\boldsymbol{X}_{(1)}), \mathrm{rank}(\boldsymbol{X}_{(2)})$ and $\mathrm{rank}(\boldsymbol{X}_{(3)})$ measure the correlation among the head entities, relations and tail entities, respectively. Entities or relations in KGs are highly correlated. For instance, some relations are mutual inverse relations or one relation may be a composition of another two relations [Zhang et al., 2021]. Thus, the overlapped trace norm $L(\boldsymbol{X};\alpha)$ can pose a constraint to the embeddings of entities and relations. Minimizing $L(\boldsymbol{X};\alpha)$ encourage a high correlation among entities and relations, which brings strong regularization and reduces overfitting. We next establish the relationship between our regularization term Eq.(3) and $L(\boldsymbol{X};\alpha)$ by Proposition 1 and Proposition 2. We will prove that Eq.(3) is an upper bound of $L(\boldsymbol{X};\alpha)$.

**Proposition 1.** *For any $\boldsymbol{X}$, and for any decomposition of $\boldsymbol{X}$, $\boldsymbol{X} = \sum_{d=1}^{D/P} \boldsymbol{W} \times_1 \boldsymbol{H}_{:d:} \times_2 \boldsymbol{R}_{:d:} \times_3 \boldsymbol{T}_{:d:}$, we have*

$$2\sqrt{\lambda_1\lambda_4}L(\boldsymbol{X};\alpha) \leq \sum_{d=1}^{D/P} \lambda_1(\|\boldsymbol{H}_{:d:}\|_F^\alpha + \|\boldsymbol{R}_{:d:}\|_F^\alpha + \|\boldsymbol{T}_{:d:}\|_F^\alpha)$$

$$+ \lambda_4(\|\boldsymbol{W} \times_2 \boldsymbol{R}_{:d:} \times_3 \boldsymbol{T}_{:d:}\|_F^\alpha + \|\boldsymbol{W} \times_3 \boldsymbol{T}_{:d:} \times_1 \boldsymbol{H}_{:d:}\|_F^\alpha + \|\boldsymbol{W} \times_1 \boldsymbol{H}_{:d:} \times_2 \boldsymbol{R}_{:d:}\|_F^\alpha) \quad (5)$$

*If $\boldsymbol{X}_{(1)} = \boldsymbol{U}_1\boldsymbol{\Sigma}_1\boldsymbol{V}_1^T, \boldsymbol{X}_{(2)} = \boldsymbol{U}_2\boldsymbol{\Sigma}_2\boldsymbol{V}_2^T, \boldsymbol{X}_{(3)} = \boldsymbol{U}_3\boldsymbol{\Sigma}_3\boldsymbol{V}_3^T$ are compact singular value decompositions of $\boldsymbol{X}_{(1)}, \boldsymbol{X}_{(2)}, \boldsymbol{X}_{(3)}$ [Bai et al., 2000], respectively, then there exists a decomposition of $\boldsymbol{X}$, $\boldsymbol{X} = \sum_{d=1}^{D/P} \boldsymbol{W} \times_1 \boldsymbol{H}_{:d:} \times_2 \boldsymbol{R}_{:d:} \times_3 \boldsymbol{T}_{:d:}$, such that the two sides of Eq.(5) equal.*

**Proposition 2.** *For any $\boldsymbol{X}$, and for any decomposition of $\boldsymbol{X}$, $\boldsymbol{X} = \sum_{d=1}^{D/P} \boldsymbol{W} \times_1 \boldsymbol{H}_{:d:} \times_2 \boldsymbol{R}_{:d:} \times_3 \boldsymbol{T}_{:d:}$, we have*

$$2\sqrt{\lambda_2\lambda_3}L(\boldsymbol{X}) \leq \sum_{d=1}^{D/P} \lambda_2(\|\boldsymbol{T}_{:d:}\|_F^\alpha\|\boldsymbol{R}_{:d:}\|_F^\alpha + \|\boldsymbol{T}_{:d:}\|_F^\alpha\|\boldsymbol{H}_{:d:}\|_F^\alpha + \|\boldsymbol{R}_{:d:}\|_F^\alpha\|\boldsymbol{H}_{:d:}\|_F^\alpha)$$

$$+ \lambda_3(\|\boldsymbol{W} \times_1 \boldsymbol{H}_{:d:}\|_F^\alpha + \|\boldsymbol{W} \times_2 \boldsymbol{R}_{:d:}\|_F^\alpha + \|\boldsymbol{W} \times_3 \boldsymbol{T}_{:d:}\|_F^\alpha) \quad (6)$$

*And there exists some $\boldsymbol{X}'$, and for any decomposition of $\boldsymbol{X}'$, such that the two sides of Eq.(6) can not achieve equality.*

Please refer to Appendix B for the proofs. Proposition 1 establishes that the r.h.s. of Eq.(5) provides a tight upper bound for $2\sqrt{\lambda_1\lambda_4}L(\boldsymbol{X};\alpha)$, while Proposition 2 demonstrates that the r.h.s. of Eq.(4) is an upper bound of $2\sqrt{\lambda_2\lambda_3}L(\boldsymbol{X};\alpha)$, but this bound is not always tight. Our proposed regularization term, Eq.(3), combines these two upper limits by adding the r.h.s. of Eq.(5) and the r.h.s. of Eq.(6). As a result, minimizing Eq.(3) can effectively minimize $L(\boldsymbol{X};\alpha)$ to regularize the model.

The two sides of Eq.(5) can achieve equality if $P = D$, meaning that the TDB model is TuckER model [Balažević et al., 2019]. Although $L(\boldsymbol{X};\alpha)$ may not always serve as a tight lower bound of the r.h.s. of Eq.(5) for TDB models other than TuckER model, it remains a common lower bound for all TDB models. To obtain a more tight lower bound, the exact values of $P$ and $\boldsymbol{W}$ are required. For example, in the case of CP model [Lacroix et al., 2018] ($P = 1$ and $\boldsymbol{W} = 1$), the nuclear 2-norm $\|\boldsymbol{X}\|_*$ is a more tight lower bound. The nuclear 2-norm is defined as follows:

$$\|\boldsymbol{X}\|_* := \min\{\sum_{d=1}^{D} \|\boldsymbol{H}_{:d1}\|_F\|\boldsymbol{R}_{:d1}\|_F\|\boldsymbol{T}_{:d1}\|_F | \boldsymbol{X} = \sum_{d=1}^{D} \boldsymbol{W} \times_1 \boldsymbol{H}_{:d1} \times_2 \boldsymbol{R}_{:d1} \times_3 \boldsymbol{T}_{:d1}\}$$

Table 1: Knowledge graph completion results on WN18RR, FB15k-237 and YGAO3-10 datasets.

| | WN18RR | | | FB15k-237 | | | YAGO3-10 | | |
|---|---|---|---|---|---|---|---|---|---|
| | MRR | H@1 | H@10 | MRR | H@1 | H@10 | MRR | H@1 | H@10 |
| CP | 0.438 | 0.416 | 0.485 | 0.332 | 0.244 | 0.507 | 0.567 | 0.495 | 0.696 |
| CP-F2 | 0.449 | 0.420 | 0.506 | 0.331 | 0.243 | 0.507 | 0.570 | 0.499 | 0.699 |
| CP-N3 | 0.469 | 0.432 | 0.541 | 0.355 | 0.261 | 0.542 | 0.575 | 0.504 | 0.703 |
| CP-DURA | 0.471 | 0.433 | 0.545 | 0.364 | 0.269 | 0.554 | 0.577 | 0.504 | 0.704 |
| CP-IVR | **0.478** | **0.437** | **0.554** | **0.365** | **0.270** | **0.555** | **0.579** | **0.507** | **0.708** |
| ComplEx | 0.464 | 0.431 | 0.526 | 0.348 | 0.256 | 0.531 | 0.574 | 0.501 | 0.704 |
| ComplEx-F2 | 0.467 | 0.431 | 0.538 | 0.349 | 0.260 | 0.529 | 0.576 | 0.502 | 0.709 |
| ComplEx-N3 | 0.491 | 0.445 | 0.578 | 0.367 | 0.272 | 0.559 | 0.577 | 0.504 | 0.707 |
| ComplEx-DURA | 0.484 | 0.440 | 0.571 | **0.372** | **0.277** | **0.563** | 0.585 | 0.512 | **0.714** |
| ComplEx-IVR | **0.494** | **0.449** | **0.581** | 0.370 | 0.275 | 0.561 | **0.586** | **0.515** | **0.714** |
| SimplE | 0.443 | 0.421 | 0.488 | 0.337 | 0.248 | 0.514 | 0.565 | 0.491 | 0.696 |
| SimplE-F2 | 0.451 | 0.422 | 0.506 | 0.338 | 0.249 | 0.514 | 0.566 | 0.494 | 0.699 |
| SimplE-IVR | **0.470** | **0.436** | **0.537** | **0.357** | **0.264** | **0.544** | **0.578** | **0.504** | **0.707** |
| ANALOGY | 0.458 | 0.424 | 0.526 | 0.348 | 0.255 | 0.530 | 0.573 | 0.502 | 0.703 |
| ANALOGY-F2 | 0.467 | 0.434 | 0.533 | 0.349 | 0.258 | 0.529 | 0.573 | 0.501 | 0.705 |
| ANALOGY-IVR | **0.482** | **0.439** | **0.568** | **0.367** | **0.272** | **0.558** | **0.582** | **0.509** | **0.713** |
| QuatE | 0.460 | 0.430 | 0.518 | 0.349 | 0.258 | 0.530 | 0.566 | 0.489 | 0.702 |
| QuatE-F2 | 0.468 | 0.435 | 0.534 | 0.349 | 0.259 | 0.529 | 0.566 | 0.489 | 0.705 |
| QuatE-IVR | **0.493** | **0.447** | **0.580** | **0.369** | **0.274** | **0.561** | **0.582** | **0.509** | **0.712** |
| TuckER | 0.446 | 0.423 | 0.490 | 0.321 | 0.233 | 0.498 | 0.551 | 0.476 | 0.689 |
| TuckER-F2 | 0.449 | 0.423 | 0.496 | 0.327 | 0.239 | 0.503 | 0.566 | 0.492 | 0.700 |
| TuckER-IVR | **0.501** | **0.460** | **0.579** | **0.368** | **0.274** | **0.555** | **0.581** | **0.508** | **0.712** |

where $\boldsymbol{W} = 1$. The following proposition establishes the relationship between $\|\boldsymbol{X}\|_*$ and $L(\boldsymbol{X}; 2)$:

**Proposition 3.** *For any $\boldsymbol{X}$, and for any decomposition of $\boldsymbol{X}$, $\boldsymbol{X} = \sum_{d=1}^{D} \boldsymbol{W} \times_1 \boldsymbol{H}_{:d1} \times_2 \boldsymbol{R}_{:d1} \times_3 \boldsymbol{T}_{:d1}$, and $\boldsymbol{W} = 1$, we have*

$$2\sqrt{\lambda_1 \lambda_4} L(\boldsymbol{X}; 2) \le 6\sqrt{\lambda_1 \lambda_4} \|\boldsymbol{X}\|_* \le \sum_{d=1}^{D} \lambda_1 (\|\boldsymbol{H}_{:d1}\|_F^2 + \|\boldsymbol{R}_{:d1}\|_F^2 + \|\boldsymbol{T}_{:d1}\|_F^{2a})$$

$$+ \lambda_4 (\|\boldsymbol{W} \times_2 \boldsymbol{R}_{:d1} \times_3 \boldsymbol{T}_{:d1}\|_F^2 + \|\boldsymbol{W} \times_3 \boldsymbol{T}_{:d1} \times_1 \boldsymbol{H}_{:d1}\|_F^2 + \|\boldsymbol{W} \times_1 \boldsymbol{H}_{:d1} \times_2 \boldsymbol{R}_{:d1}\|_F^2)$$

Although the r.h.s. of Eq.(6) is not always a tight upper bound for $2\sqrt{\lambda_2 \lambda_3} L(\boldsymbol{X}; \alpha)$ like the r.h.s. of Eq.(5), we observe that minimizing the combination of these two bounds, Eq.(3), can lead to better performance. The reason behind this is that the r.h.s. of Eq.(5) is neither an upper bound nor a lower bound of the r.h.s. of Eq.(6) for all $\boldsymbol{X}$. We present Proposition 4 in Appendix B to prove this claim.

## 4 Experiments

We first introduce the experimental settings in Section 4.1 and show the results in Section 4.2. We next conduct ablation studies in Section 4.3. Finally, we verify the reliability of our proposed upper bounds in Section 4.4. Please refer to Appendix C for more experimental details.

### 4.1 Experimental Settings

**Datasets** We evaluate the models on three KGC datasets, WN18RR [Dettmers et al., 2018], FB15k-237 [Toutanova et al., 2015] and YAGO3-10 [Dettmers et al., 2018].

**Models** We use CP, ComplEx, SimplE, ANALOGY, QuatE and TuckER as baselines. We denote CP with squared Frobenius norm method [Yang et al., 2014] as CP-F2, CP with N3 method [Lacroix

Table 2: The results on WN18RR and FB15k-237 datasets with different upper bounds.

| | WN18RR | | | FB15k-237 | | | YAGO3-10 | | |
|---|---|---|---|---|---|---|---|---|---|
| | MRR | H@1 | H@10 | MRR | H@1 | H@10 | MRR | H@1 | H@10 |
| TuckER | 0.446 | 0.423 | 0.490 | 0.321 | 0.233 | 0.498 | 0.551 | 0.476 | 0.689 |
| TuckER-IVR-1 | 0.497 | 0.455 | 0.578 | 0.366 | 0.272 | 0.553 | 0.576 | 0.501 | 0.709 |
| TuckER-IVR-2 | 0.459 | 0.423 | 0.518 | 0.336 | 0.241 | 0.518 | 0.568 | 0.493 | 0.700 |
| TuckER-IVR | **0.501** | **0.460** | **0.579** | **0.368** | **0.274** | **0.555** | **0.581** | **0.508** | **0.712** |

Table 3: The results on Kinship dataset with different upper bounds.

| | $\|\boldsymbol{X}_{(1)}\|_*$ | $\|\boldsymbol{X}_{(2)}\|_*$ | $\|\boldsymbol{X}_{(3)}\|_*$ | $L(\boldsymbol{X})$ |
|---|---|---|---|---|
| TuckER | 10,719 | 8,713 | 11,354 | 30,786 |
| TuckER-IVR-1 | 5,711 | 5,021 | 6,271 | 17,003 |
| TuckER-IVR-2 | 6,145 | 5,441 | 6,744 | 18,330 |
| TuckER-IVR | **3,538** | **3,511** | **3,988** | **11,037** |

et al., 2018] as CP-N3, CP with DURA method [Zhang et al., 2020] as CP-DURA and CP with IVR method as CP-IVR. The notations for other models are similar to the notations for CP.

**Evaluation Metrics**  We use the filtered MRR and Hits@N (H@N) [Bordes et al., 2013] as evaluation metrics and choose the hyper-parameters with the best filtered MRR on the validation set. We run each model three times with different random seeds and report the mean results.

## 4.2  Results

See Table 1 for the results. For CP and ComplEx, the models that N3 and DURA are suitable, the results show that N3 enhances the models more than F2, and DURA outperforms both F2 and N3, leading to substantial improvements. IVR achieves better performance than DURA on WN18RR dataset and achieves similar performance to DURA on FB15k-237 and YAGO3-10 dataset. For SimplE, ANALOGY, QuatE, and TuckER, the improvement offered by F2 is minimal, while IVR significantly boosts model performance. In summary, these results demonstrate the effectiveness and generality of IVR.

## 4.3  Ablation Studies

We conduct ablation studies to examine the effectiveness of the upper bounds. Our notations for the models are as follows: the model with upper bound Eq.(5) is denoted as IVR-1, model with upper bound Eq.(6) as IVR-2, and model with upper bound Eq.(3) as IVR.

See Table 2 for the results. We use TuckER as the baseline. IVR with only 1 regularization coefficient, IVR-1, achieves comparable results with vanilla IVR, which shows that IVR can still perform well with fewer hyper-parameters. IVR-1 outperforms IVR-2 due to the tightness of Eq.(5).

## 4.4  Upper Bounds

We verify that minimizing the upper bounds can effectively minimize $L(\boldsymbol{X}; \alpha)$. $L(\boldsymbol{X}; \alpha)$ can measure the correlation of $\boldsymbol{X}$. Lower values of $L(X; \alpha)$ encourage higher correlations among entities and relations, and thus bring a strong constraint for regularization. Upon training the models, we compute $L(\boldsymbol{X}; \alpha)$. As computing $L(\boldsymbol{X}; \alpha)$ for large KGs is impractical, we conduct experiments on a small KG dataset, Kinship [Kok and Domingos, 2007], which consists of 104 entities and 25 relations. We use TuckER as the baseline and compare it against IVR-1 (Eq.(5)), IVR-2 (Eq.(6)), and IVR (Eq.(3)).

See Table 3 for the results. Our results demonstrate that the upper bounds are effective in minimizing $L(\boldsymbol{X}; \alpha)$. All three upper bounds can lead to a decrease of $L(\boldsymbol{X}; \alpha)$, achieving better performance (Table 2) by more effective regularization. The $L(\boldsymbol{X}; \alpha)$ of IVR-1 is smaller than that of IVR-2

because the upper bound in Eq.(5) is tight. IVR, which combines IVR-1 and IVR-2, produces the most reduction of $L(\boldsymbol{X}; \alpha)$. This finding suggests that combining the two upper bounds can be more effective. Overall, our experimental results confirm the reliability of our theoretical analysis.

## 5 Conclusion

In this paper, we undertake an analysis of TDB models in KGC. We first offer a summary of TDB models and derive a general form that facilitates further analysis. TDB models often suffer from overfitting, and thus, we propose a regularization based on our derived general form. It is applicable to most TDB models and improve the model performance. We further propose a theoretical analysis to support our regularization and experimentally validate our theoretical analysis. Our regularization is limited to TDB models, hoping that more regularization will be proposed in other types of models, such as translation-based models and neural networks models. We also intend to explore how to apply our regularization to other fields, such as tensor completion [Song et al., 2019].

## Acknowledgements

This work is supported by the National Key Research and Development Program of China (2020AAA0106000), the National Natural Science Foundation of China (U19A2079), and the CCCD Key Lab of Ministry of Culture and Tourism.

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

# A Logical Rules

KGs often involve some logical rules to capture inductive capacity [Zhang et al., 2021]. Thus, we analyze how to design models such that the models can learn the symmetry, antisymmetry and inverse rules. We have derived a general form for TDB models, Eq.(1). Next, we study how to enable Eq.(1) to learn logical rules. We first define the symmetry rules, antisymmetry rules and inverse rules. We denote $\boldsymbol{X}_{ijk} = f(\boldsymbol{H}_{i::}, \boldsymbol{R}_{j::}, \boldsymbol{T}_{k::})$ as $f(\boldsymbol{h}, \boldsymbol{r}, \boldsymbol{t})$ for simplicity. We assume $\boldsymbol{H} = \boldsymbol{T}$, which mainly aims to reduce overfitting Yang et al. [2014]. Existing TDB models for KGC except CP [Lacroix et al., 2018] all forces $\boldsymbol{H} = \boldsymbol{T}$.

A relation $r$ is symmetric if $\forall h, t, (h, r, t) \in \mathcal{S} \rightarrow (t, r, h) \in \mathcal{S}$. A model is able to learn the symmetry rules if

$$\exists \boldsymbol{r} \in \mathbb{R}^D \wedge \boldsymbol{r} \neq 0, \forall \boldsymbol{h}, \boldsymbol{t} \in \mathbb{R}^D, f(\boldsymbol{h}, \boldsymbol{r}, \boldsymbol{t}) = f(\boldsymbol{t}, \boldsymbol{r}, \boldsymbol{h})$$

A relation $r$ is antisymmetric if $\forall h, t, (h, r, t) \in \mathcal{S} \rightarrow (t, r, h) \notin \mathcal{S}$. A model is able to learn the antisymmetry rules if

$$\exists \boldsymbol{r} \in \mathbb{R}^D \wedge \boldsymbol{r} \neq 0, \forall \boldsymbol{h}, \boldsymbol{t} \in \mathbb{R}^D, f(\boldsymbol{h}, \boldsymbol{r}, \boldsymbol{t}) = -f(\boldsymbol{t}, \boldsymbol{r}, \boldsymbol{h})$$

A relation $r_1$ is inverse to a relation $r_2$ if $\forall h, t, (h, r_1, t) \in \mathcal{S} \rightarrow (t, r_2, h) \in \mathcal{S}$. A model is able to learn the inverse rules if

$$\forall \boldsymbol{r}_1 \in \mathbb{R}^D, \exists \boldsymbol{r}_2 \in \mathbb{R}^D, \forall \boldsymbol{h}, \boldsymbol{t} \in \mathbb{R}^D, f(\boldsymbol{h}, \boldsymbol{r}_1, \boldsymbol{t}) = f(\boldsymbol{t}, \boldsymbol{r}_2, \boldsymbol{h})$$

We restrict $\boldsymbol{r} \neq 0$ because $\boldsymbol{r} = 0$ will result in $f$ equal to an identically zero function. By choosing different $P$ and $\boldsymbol{W}$, we can define different TDB models as discussed in Section 3.1. Next, we give a theoretical analysis to establish the relationship between logical rules and TDB models.

**Theorem 1.** *Assume a model can be represented as the form of Eq.(1), then a model is able to learn the symmetry rules iff* $\operatorname{rank}(\boldsymbol{W}_{(2)}^T - \boldsymbol{S}\boldsymbol{W}_{(2)}^T) < P$. *A model is able to learn the antisymmetry rules iff* $\operatorname{rank}(\boldsymbol{W}_{(2)}^T + \boldsymbol{S}\boldsymbol{W}_{(2)}^T) < P$. *A model is able to learn the inverse rules iff* $\operatorname{rank}(\boldsymbol{W}_{(2)}^T) = \operatorname{rank}([\boldsymbol{W}_{(2)}^T, \boldsymbol{S}\boldsymbol{W}_{(2)}^T])$, *where* $\boldsymbol{S} \in \mathbb{R}^{P^2 \times P^2}$ *is a permutation matrix with* $\boldsymbol{S}_{(i-1)P+j,(j-1)P+i} = 1 (i, j = 1, 2, \ldots, P)$ *and otherwise 0,* $[\boldsymbol{W}_{(2)}^T, \boldsymbol{S}\boldsymbol{W}_{(2)}^T]$ *is the concatenation of matrix* $\boldsymbol{W}_{(2)}^T$ *and matrix* $\boldsymbol{S}\boldsymbol{W}_{(2)}^T$.

*Proof.* According to the symmetry rules, for any triplet $(i, j, k)$, we have that $f(i, j, k) = f(k, j, i)$, i.e., $f(\mathbf{H}_{i:}, \mathbf{R}_{j:}, \mathbf{T}_{k:}) = f(\mathbf{H}_{k:}, \mathbf{R}_{j:}, \mathbf{T}_{i:}) = f(\mathbf{T}_{k:}, \mathbf{R}_{j:}, \mathbf{H}_{i:})$ (we use $\mathbf{H} = \mathbf{T}$ here). We replace $\mathbf{H}_{i:}$ by $\mathbf{h}$, replace $\mathbf{R}_{j:}$ by $\mathbf{r}$, replace $\mathbf{T}_{k:}$ by $\mathbf{t}$, then we have $f(\mathbf{h}, \mathbf{r}, \mathbf{t}) = f(\mathbf{t}, \mathbf{r}, \mathbf{h})$.

Then we have that

$$\sum_{l=1}^{P} \sum_{m=1}^{P} \sum_{n=1}^{P} \boldsymbol{W}_{lmn} \langle \boldsymbol{h}_{:l}, \boldsymbol{r}_{:m}, \boldsymbol{t}_{:n} \rangle - \sum_{l=1}^{P} \sum_{m=1}^{P} \sum_{n=1}^{P} \boldsymbol{W}_{lmn} \langle \boldsymbol{t}_{:l}, \boldsymbol{r}_{:m}, \boldsymbol{h}_{:n} \rangle$$

$$= \sum_{l=1}^{P} \sum_{m=1}^{P} \sum_{n=1}^{P} \boldsymbol{W}_{lmn} \langle \boldsymbol{h}_{:l}, \boldsymbol{r}_{:m}, \boldsymbol{t}_{:n} \rangle - \sum_{l=1}^{P} \sum_{m=1}^{P} \sum_{n=1}^{P} \boldsymbol{W}_{nml} \langle \boldsymbol{h}_{:l}, \boldsymbol{r}_{:m}, \boldsymbol{t}_{:n} \rangle$$

$$= \sum_{l=1}^{P} \sum_{m=1}^{P} \sum_{n=1}^{P} (\boldsymbol{W}_{lmn} - \boldsymbol{W}_{nml}) \langle \boldsymbol{h}_{:l}, \boldsymbol{r}_{:m}, \boldsymbol{t}_{:n} \rangle$$

$$= \sum_{l=1}^{P} \sum_{m=1}^{P} \sum_{n=1}^{P} (\boldsymbol{W}_{lmn} - \boldsymbol{W}_{nml}) \boldsymbol{r}_{:m}^T (\boldsymbol{h}_{:l} * \boldsymbol{t}_{:n})$$

$$= \sum_{l=1}^{P} \sum_{n=1}^{P} (\sum_{m=1}^{P} (\boldsymbol{W}_{lmn} - \boldsymbol{W}_{nml}) \boldsymbol{r}_{:m}^T)(\boldsymbol{h}_{:l} * \boldsymbol{t}_{:n}) = 0$$

where $*$ is the Hadamard product. Since the above equation holds for any $\boldsymbol{h}, \boldsymbol{t} \in \mathbb{R}^D$, we can get

$$\sum_{m=1}^{P} (\boldsymbol{W}_{lmn} - \boldsymbol{W}_{nml}) \boldsymbol{r}_{:m}^T = \boldsymbol{0} (l, n = 1, 2, \ldots, P)$$

Therefore, a model is able to learn the symmetry rules iff

$$\exists \boldsymbol{r} \in \mathbb{R}^D \wedge \boldsymbol{r} \neq 0, \sum_{m=1}^{P} (\boldsymbol{W}_{lmn} - \boldsymbol{W}_{nml})\boldsymbol{r}_{:m}^T = \boldsymbol{0}$$

Therefore, the symmetry rule is transformed into a system of linear equations. This system of linear equations have non-zero solution iff $\mathrm{rank}(\boldsymbol{W}_{(2)}^T - \boldsymbol{S}\boldsymbol{W}_{(2)}^T) < P$. Thus, a model is able to learn the symmetry rules iff $\mathrm{rank}(\boldsymbol{W}_{(2)}^T - \boldsymbol{S}\boldsymbol{W}_{(2)}^T) < P$.

Similarly, a model is able to learn the anti-symmetry rules iff $\mathrm{rank}(\boldsymbol{W}_{(2)}^T + \boldsymbol{S}\boldsymbol{W}_{(2)}^T) < P$.

For the inverse rule:

$$\forall \boldsymbol{r}_1 \in \mathbb{R}^D, \exists \boldsymbol{r}_2 \in \mathbb{R}^D, \forall \boldsymbol{h}, \boldsymbol{t} \in \mathbb{R}^D, f(\boldsymbol{h}, \boldsymbol{r}_1, \boldsymbol{t}) = f(\boldsymbol{t}, \boldsymbol{r}_2, \boldsymbol{h})$$

a model is able to learn the symmetry rules iff the following equation

$$\boldsymbol{W}_{(2)}^T \boldsymbol{r}_1 = \boldsymbol{S}\boldsymbol{W}_{(2)}^T \boldsymbol{r}_2$$

for any $\boldsymbol{r}_1 \in \mathbb{R}^D$, there exists $\boldsymbol{r}_2 \in \mathbb{R}^D$ such that the equation holds. Thus, the column vectors of $\boldsymbol{W}_{(2)}^T$ can be expressed linearly by the column vectors of $\boldsymbol{S}\boldsymbol{W}_{(2)}^T$. Since $\boldsymbol{S}$ is a permutation matrix and $\boldsymbol{S} = \boldsymbol{S}^T$, we have that $\boldsymbol{S}^2 = \boldsymbol{I}$, thus

$$\boldsymbol{S}\boldsymbol{W}_{(2)}^T \boldsymbol{r}_1 = \boldsymbol{S}^2 \boldsymbol{W}_{(2)}^T \boldsymbol{r}_2 = \boldsymbol{W}_{(2)}^T \boldsymbol{r}_2$$

For any $\boldsymbol{r}_1 \in \mathbb{R}^D$, there exists $\boldsymbol{r}_2 \in \mathbb{R}^D$ such that the above equation holds. Thus, the column vectors of $\boldsymbol{S}\boldsymbol{W}_{(2)}^T$ can be expressed linearly by the column vectors of $\boldsymbol{W}_{(2)}^T$. Therefore, the column space of $\boldsymbol{W}_{(2)}^T$ is equivalent to the column space of $\boldsymbol{S}\boldsymbol{W}_{(2)}^T$, thus we have

$$\mathrm{rank}(\boldsymbol{W}_{(2)}^T) = \mathrm{rank}(\boldsymbol{S}\boldsymbol{W}_{(2)}^T) = \mathrm{rank}([\boldsymbol{W}_{(2)}^T, \boldsymbol{S}\boldsymbol{W}_{(2)}^T])$$

Meanwhile, if $\mathrm{rank}(\boldsymbol{W}_{(2)}^T) = \mathrm{rank}([\boldsymbol{W}_{(2)}^T, \boldsymbol{S}\boldsymbol{W}_{(2)}^T])$, then the columns of $\boldsymbol{W}_{(2)}^T$ can be expressed linearly by the columns of $\boldsymbol{S}\boldsymbol{W}_{(2)}^T$, thus

$$\forall \boldsymbol{r}_1 \in \mathbb{R}^D, \exists \boldsymbol{r}_2 \in \mathbb{R}^D, \boldsymbol{W}_{(2)}^T \boldsymbol{r}_1 = \boldsymbol{S}\boldsymbol{W}_{(2)}^T \boldsymbol{r}_2$$

Thus, a model is able to learn the inverse rules iff $\mathrm{rank}(\boldsymbol{W}_{(2)}^T) = \mathrm{rank}([\boldsymbol{W}_{(2)}^T, \boldsymbol{S}\boldsymbol{W}_{(2)}^T])$. $\qquad\square$

By this theoerm, we only need to judge the relationship between $P$ and the matrix rank about $\boldsymbol{W}_{(2)}$. ComplEx, SimplE, ANALOGYY and QuatE design specific core tensors to make the models enable to learn the logical rules. We can easily verify that these models satisfy the conditions in this theorem. For example, for ComplEx, we have that $P = 2$ and

$$\boldsymbol{W}_{(2)}^T = \begin{pmatrix} 1 & 0 \\ 0 & 1 \\ 0 & -1 \\ 1 & 0 \end{pmatrix}, \boldsymbol{S}\boldsymbol{W}_{(2)}^T = \begin{pmatrix} 1 & 0 \\ 0 & -1 \\ 0 & 1 \\ 1 & 0 \end{pmatrix}, [\boldsymbol{W}_{(2)}, \boldsymbol{S}\boldsymbol{W}_{(2)}^T] = \begin{pmatrix} 1 & 0 & 1 & 0 \\ 0 & 1 & 0 & -1 \\ 0 & -1 & 0 & 1 \\ 1 & 0 & 1 & 0 \end{pmatrix}$$

$$\mathrm{rank}(\boldsymbol{W}_{(2)}^T - \boldsymbol{S}\boldsymbol{W}_{(2)}^T) = \mathrm{rank}(\begin{pmatrix} 0 & 0 \\ 0 & 2 \\ 0 & -2 \\ 0 & 0 \end{pmatrix}) = 1 < P = 2$$

$$\mathrm{rank}(\boldsymbol{W}_{(2)}^T + \boldsymbol{S}\boldsymbol{W}_{(2)}^T) = \mathrm{rank}(\begin{pmatrix} 2 & 0 \\ 0 & 0 \\ 0 & 0 \\ 2 & 0 \end{pmatrix}) = 1 < P = 2$$

$$\mathrm{rank}(\boldsymbol{W}_{(2)}^T) = \mathrm{rank}([\boldsymbol{W}_{(2)}, \boldsymbol{S}\boldsymbol{W}_{(2)}^T]) = 2$$

Thus, ComplEx is able to learn the symmetry rules, antisymmetry rules and inverse rules.

# B Proofs

To prove the Proposition 1 and Proposition 2, we first prove the following lemma.

**Lemma 1.**

$$\|\boldsymbol{Z}\|_*^\alpha = \min_{\boldsymbol{Z}=\boldsymbol{U}\boldsymbol{V}^T} \frac{1}{2}(\lambda\|\boldsymbol{U}\|_F^{2\alpha} + \frac{1}{\lambda}\|\boldsymbol{V}\|_F^{2\alpha})$$

*where $\alpha > 0$, $\lambda > 0$ and $\{\boldsymbol{Z}, \boldsymbol{U}, \boldsymbol{V}\}$ are real matrices. If $\boldsymbol{Z} = \hat{\boldsymbol{U}}\boldsymbol{\Sigma}\hat{\boldsymbol{V}}^T$ is a singular value decomposition of $\boldsymbol{Z}$, then equality holds for the choice $\boldsymbol{U} = \lambda^{\frac{-1}{2\alpha}}\hat{\boldsymbol{U}}\sqrt{\boldsymbol{\Sigma}}$ and $\boldsymbol{V} = \lambda^{\frac{1}{2\alpha}}\hat{\boldsymbol{V}}\sqrt{\boldsymbol{\Sigma}}$, where $\sqrt{\boldsymbol{\Sigma}}$ is the element-wise square root of $\Sigma$.*

*Proof.* The proof of Lemma 1 is based on the proof of Lemma 9 in [Ciliberto et al., 2017]. Let the singular value decomposition of $\boldsymbol{Z} \in \mathbb{R}^{m \times n}$ be $\boldsymbol{Z} = \hat{\boldsymbol{U}}\boldsymbol{\Sigma}\hat{\boldsymbol{V}}^T$, where $\hat{\boldsymbol{U}} \in \mathbb{R}^{m \times r}, \boldsymbol{\Sigma} \in \mathbb{R}^{r \times r}, \hat{\boldsymbol{V}} \in \mathbb{R}^{n \times r}, r = \text{rank}(\boldsymbol{Z}), \hat{\boldsymbol{U}}^T\hat{\boldsymbol{U}} = \boldsymbol{I}_{r \times r}$ and $\hat{\boldsymbol{V}}^T\hat{\boldsymbol{V}} = \boldsymbol{I}_{r \times r}$. We choose any $\boldsymbol{U}, \boldsymbol{V}$ such that $\boldsymbol{Z} = \boldsymbol{U}\boldsymbol{V}^T$, then we have $\boldsymbol{\Sigma} = \hat{\boldsymbol{U}}^T\boldsymbol{U}\boldsymbol{V}^T\hat{\boldsymbol{V}}$. Moreover, since $\hat{\boldsymbol{U}}, \hat{\boldsymbol{V}}$ have orthogonal columns, $\|\hat{\boldsymbol{U}}^T\boldsymbol{U}\|_F \leq \|\boldsymbol{U}\|_F, \|\hat{\boldsymbol{V}}^T\boldsymbol{V}\|_F \leq \|\boldsymbol{V}\|_F$. Then

$$\|\boldsymbol{Z}\|_*^\alpha = \text{Tr}(\boldsymbol{\Sigma})^\alpha = \text{Tr}(\hat{\boldsymbol{U}}^T\boldsymbol{U}\boldsymbol{V}^T\hat{\boldsymbol{V}})^\alpha \leq \|\hat{\boldsymbol{U}}^T\boldsymbol{U}\|_F^\alpha\|\boldsymbol{V}\hat{\boldsymbol{V}}^T\|_F^\alpha$$

$$\leq (\sqrt{\lambda}\|\boldsymbol{U}\|_F^\alpha)(\frac{1}{\sqrt{\lambda}}\|\boldsymbol{V}\|_F^\alpha) \leq \frac{1}{2}(\lambda\|\boldsymbol{U}\|_F^{2\alpha} + \frac{1}{\lambda}\|\boldsymbol{V}\|_F^{2\alpha})$$

where the first upper bound is Cauchy-Schwarz inequality and the third upper bound is AM-GM inequality.

Let $\boldsymbol{U} = \lambda^{\frac{-1}{2\alpha}}\hat{\boldsymbol{U}}\sqrt{\boldsymbol{\Sigma}}$ and $\boldsymbol{V} = \lambda^{\frac{1}{2\alpha}}\hat{\boldsymbol{V}}\sqrt{\boldsymbol{\Sigma}}$, we have that

$$\frac{1}{2}(\lambda\|\boldsymbol{U}\|_F^{2\alpha} + \frac{1}{\lambda}\|\boldsymbol{V}\|_F^{2\alpha}) = \frac{1}{2}(\|\hat{\boldsymbol{U}}\sqrt{\boldsymbol{\Sigma}}\|_F^{2\alpha} + \|\hat{\boldsymbol{V}}\sqrt{\boldsymbol{\Sigma}}\|_F^{2\alpha}) = \frac{1}{2}(\|\sqrt{\boldsymbol{\Sigma}}\|_F^{2\alpha} + \|\sqrt{\boldsymbol{\Sigma}}\|_F^{2\alpha}) = \|\boldsymbol{Z}\|_*^\alpha$$

In summary,

$$\|\boldsymbol{Z}\|_*^\alpha = \min_{\boldsymbol{z}=\boldsymbol{U}\boldsymbol{V}^T} \frac{1}{2}(\lambda\|\boldsymbol{U}\|_F^{2\alpha} + \frac{1}{\lambda}\|\boldsymbol{V}\|_F^{2\alpha})$$

$\square$

**Proposition 1.** *For any $\boldsymbol{X}$, and for any decomposition of $\boldsymbol{X}$, $\boldsymbol{X} = \sum_{d=1}^{D/P} \boldsymbol{W} \times_1 \boldsymbol{H}_{:d:} \times_2 \boldsymbol{R}_{:d:} \times_3 \boldsymbol{T}_{:d:}$, we have*

$$2\sqrt{\lambda_1\lambda_4}L(\boldsymbol{X};\alpha) \leq \sum_{d=1}^{D/P} \lambda_1(\|\boldsymbol{H}_{:d:}\|_F^\alpha + \|\boldsymbol{R}_{:d:}\|_F^\alpha + \|\boldsymbol{T}_{:d:}\|_F^\alpha)$$

$$+ \lambda_4(\|\boldsymbol{W} \times_2 \boldsymbol{R}_{:d:} \times_3 \boldsymbol{T}_{:d:}\|_F^\alpha + \|\boldsymbol{W} \times_3 \boldsymbol{T}_{:d:} \times_1 \boldsymbol{H}_{:d:}\|_F^\alpha + \|\boldsymbol{W} \times_1 \boldsymbol{H}_{:d:} \times_2 \boldsymbol{R}_{:d:}\|_F^\alpha) \quad (7)$$

*If $\boldsymbol{X}_{(1)} = \boldsymbol{U}_1\boldsymbol{\Sigma}_1\boldsymbol{V}_1^T, \boldsymbol{X}_{(2)} = \boldsymbol{U}_2\boldsymbol{\Sigma}_2\boldsymbol{V}_2^T, \boldsymbol{X}_{(3)} = \boldsymbol{U}_3\boldsymbol{\Sigma}_3\boldsymbol{V}_3^T$ are compact singular value decompositions of $\boldsymbol{X}_{(1)}, \boldsymbol{X}_{(2)}, \boldsymbol{X}_{(3)}$, respectively, then there exists a decomposition of $\boldsymbol{X}$, $\boldsymbol{X} = \sum_{d=1}^{D/P} \boldsymbol{W} \times_1 \boldsymbol{H}_{:d:} \times_2 \boldsymbol{R}_{:d:} \times_3 \boldsymbol{T}_{:d:}$, such that the two sides of Eq.(5) equal, where $P = D$, $\boldsymbol{H}_{:1:} = \sqrt{\lambda_1/\lambda_4}^{\frac{-1}{\alpha}}\boldsymbol{U}_1\sqrt{\boldsymbol{\Sigma}_1}, \boldsymbol{R}_{:1:} = \sqrt{\lambda_1/\lambda_4}^{\frac{-1}{\alpha}}\boldsymbol{U}_2\sqrt{\boldsymbol{\Sigma}_2}, \boldsymbol{T}_{:1:} = \sqrt{\lambda_1/\lambda_4}^{\frac{-1}{\alpha}}\boldsymbol{U}_3\sqrt{\boldsymbol{\Sigma}_3}, \boldsymbol{W} = \sqrt{\lambda_1/\lambda_4}^{\frac{3}{\alpha}}\boldsymbol{X} \times_1 \sqrt{\boldsymbol{\Sigma}_1^{-1}}\boldsymbol{U}_1^T \times_2 \sqrt{\boldsymbol{\Sigma}_2^{-1}}\boldsymbol{U}_2^T \times_3 \sqrt{\boldsymbol{\Sigma}_3^{-1}}\boldsymbol{U}_3^T$.*

*Proof.* Let the $n$-rank [Kolda and Bader, 2009] of $\boldsymbol{X} \in \mathbb{R}^{n_1 \times n_2 \times n_3}$ be $(r_1, r_2, r_3)$, then $\boldsymbol{U}_1 \in \mathbb{R}^{n_1 \times r_1}, \boldsymbol{U}_2 \in \mathbb{R}^{n_2 \times r_2}, \boldsymbol{U}_3 \in \mathbb{R}^{n_3 \times r_3}, \boldsymbol{\Sigma}_1 \in \mathbb{R}^{r_1 \times r_1}, \boldsymbol{\Sigma}_2 \in \mathbb{R}^{r_2 \times r_2}, \boldsymbol{\Sigma}_3 \in \mathbb{R}^{r_3 \times r_3}, \boldsymbol{W} \in \mathbb{R}^{r_1 \times r_2 \times r_3}$.

If $\boldsymbol{X} = \boldsymbol{0}$, the above proposition is obviously true, we define $\boldsymbol{0}^{-1} := \boldsymbol{0}$ here.

For any $\boldsymbol{X} \neq \boldsymbol{0}$, since $\boldsymbol{X}_{(1)} = \sum_{d=1}^{D/P} \boldsymbol{H}_{:d:}(\boldsymbol{W}_{(1)}(\boldsymbol{T}_{:d:} \otimes \boldsymbol{R}_{:d:})^T), \boldsymbol{X}_{(2)} = \sum_{d=1}^{D/P} \boldsymbol{R}_{:d:}(\boldsymbol{W}_{(2)}(\boldsymbol{T}_{:d:} \otimes \boldsymbol{H}_{:d:})^T), \boldsymbol{X}_{(3)} = \sum_{d=1}^{D/P} \boldsymbol{T}_{:d:}(\boldsymbol{W}_{(3)}(\boldsymbol{R}_{:d:} \otimes \boldsymbol{H}_{:d:})^T)$, by applying Lemma 1 to $\boldsymbol{X}_{(1)}, \boldsymbol{X}_{(2)}, \boldsymbol{X}_{(3)}$, we

have that

$$2L(\boldsymbol{X};\alpha)$$

$$\leq \sum_{d=1}^{D/P} \|\boldsymbol{H}_{:d:}(\boldsymbol{W}_{(1)}(\boldsymbol{T}_{:d:} \otimes \boldsymbol{R}_{:d:})^T)\|_*^{\alpha/2} + \|\boldsymbol{R}_{:d:}(\boldsymbol{W}_{(2)}(\boldsymbol{T}_{:d:} \otimes \boldsymbol{H}_{:d:})^T)\|_*^{\alpha/2} + \|\boldsymbol{T}_{:d:}(\boldsymbol{W}_{(3)}(\boldsymbol{R}_{:d:} \otimes \boldsymbol{H}_{:d:})^T)\|_*^{\alpha/2}$$

$$\leq \sum_{d=1}^{D/P} \lambda(\|\boldsymbol{H}_{:d:}\|_F^\alpha + \|\boldsymbol{R}_{:d:}\|_F^\alpha + \|\boldsymbol{T}_{:d:}\|_F^\alpha)$$

$$+ \frac{1}{\lambda}(\|\boldsymbol{W}_{(1)}(\boldsymbol{T}_{:d:} \otimes \boldsymbol{R}_{:d:})^T\|_F^\alpha + \|\boldsymbol{W}_{(2)}(\boldsymbol{T}_{:d:} \otimes \boldsymbol{H}_{:d:})^T\|_F^\alpha + \|\boldsymbol{W}_{(3)}(\boldsymbol{R}_{:d:} \otimes \boldsymbol{H}_{:d:})^T\|_F^\alpha)$$

$$= \sum_{d=1}^{D/P} \lambda(\|\boldsymbol{H}_{:d:}\|_F^\alpha + \|\boldsymbol{R}_{:d:}\|_F^\alpha + \|\boldsymbol{T}_{:d:}\|_F^\alpha)$$

$$+ \frac{1}{\lambda}(\|\boldsymbol{W} \times_2 \boldsymbol{R}_{:d:} \times_3 \boldsymbol{T}_{:d:}\|_F^\alpha + \|\boldsymbol{W} \times_3 \boldsymbol{T}_{:d:} \times_1 \boldsymbol{H}_{:d:}\|_F^\alpha + \|\boldsymbol{W} \times_1 \boldsymbol{H}_{:d:} \times_2 \boldsymbol{R}_{:d:}\|_F^\alpha)$$

Let $\lambda = \sqrt{\lambda_1/\lambda_4}$, we have that

$$2\sqrt{\lambda_1\lambda_4}L(\boldsymbol{X};\alpha) \leq \sum_{d=1}^{D/P} \lambda_1(\|\boldsymbol{H}_{:d:}\|_F^\alpha + \|\boldsymbol{R}_{:d:}\|_F^\alpha + \|\boldsymbol{T}_{:d:}\|_F^\alpha)$$

$$+ \lambda_4(\|\boldsymbol{W} \times_2 \boldsymbol{R}_{:d:} \times_3 \boldsymbol{T}_{:d:}\|_F^\alpha + \|\boldsymbol{W} \times_3 \boldsymbol{T}_{:d:} \times_1 \boldsymbol{H}_{:d:}\|_F^\alpha + \|\boldsymbol{W} \times_1 \boldsymbol{H}_{:d:} \times_2 \boldsymbol{R}_{:d:}\|_F^\alpha)$$

Since $\boldsymbol{U}_1^T\boldsymbol{U}_1 = \boldsymbol{I}_{r_1 \times r_1}, \boldsymbol{U}_2^T\boldsymbol{U}_2 = \boldsymbol{I}_{r_2 \times r_2}, \boldsymbol{U}_3^T\boldsymbol{U}_3 = \boldsymbol{I}_{r_3 \times r_3}$, thus we have $\boldsymbol{X}_{(1)} = \boldsymbol{U}_1\boldsymbol{U}_1^T\boldsymbol{X}_{(1)}, \boldsymbol{X}_{(2)} = \boldsymbol{U}_2\boldsymbol{U}_2^T\boldsymbol{X}_{(2)}, \boldsymbol{X}_{(3)} = \boldsymbol{U}_3\boldsymbol{U}_3^T\boldsymbol{X}_{(3)}$, then

$$\boldsymbol{X} = \boldsymbol{X} \times_1 (\boldsymbol{U}_1\boldsymbol{U}_1^T) \times_2 (\boldsymbol{U}_2\boldsymbol{U}_2^T) \times_3 (\boldsymbol{U}_3\boldsymbol{U}_3^T)$$

$$= \boldsymbol{X} \times_1 (\boldsymbol{U}_1\sqrt{\boldsymbol{\Sigma}_1}\sqrt{\boldsymbol{\Sigma}_1^{-1}}\boldsymbol{U}_1^T) \times_2 (\boldsymbol{U}_2\sqrt{\boldsymbol{\Sigma}_2}\sqrt{\boldsymbol{\Sigma}_2^{-1}}\boldsymbol{U}_2^T) \times_3 (\boldsymbol{U}_3\sqrt{\boldsymbol{\Sigma}_3}\sqrt{\boldsymbol{\Sigma}_3^{-1}}\boldsymbol{U}_3^T)$$

$$= \boldsymbol{X} \times_1 \sqrt{\boldsymbol{\Sigma}_1^{-1}}\boldsymbol{U}_1^T \times_2 \sqrt{\boldsymbol{\Sigma}_2^{-1}}\boldsymbol{U}_2^T \times_3 \sqrt{\boldsymbol{\Sigma}_3^{-1}}\boldsymbol{U}_3^T \times_1 \boldsymbol{U}_1\sqrt{\boldsymbol{\Sigma}_1} \times_2 \boldsymbol{U}_2\sqrt{\boldsymbol{\Sigma}_2} \times_3 \boldsymbol{U}_3\sqrt{\boldsymbol{\Sigma}_3}$$

Let $P = D$ and $\boldsymbol{H}_{:1:} = \sqrt{\lambda_1/\lambda_4}^{\frac{-1}{\alpha}}\boldsymbol{U}_1\sqrt{\boldsymbol{\Sigma}_1}, \boldsymbol{R}_{:1:} = \sqrt{\lambda_1/\lambda_4}^{\frac{-1}{\alpha}}\boldsymbol{U}_2\sqrt{\boldsymbol{\Sigma}_2}, \boldsymbol{T}_{:1:} = \sqrt{\lambda_1/\lambda_4}^{\frac{-1}{\alpha}}\boldsymbol{U}_3\sqrt{\boldsymbol{\Sigma}_3}, \boldsymbol{W} = \sqrt{\lambda_1/\lambda_4}^{\frac{3}{\alpha}}\boldsymbol{X} \times_1 \sqrt{\boldsymbol{\Sigma}_1^{-1}}\boldsymbol{U}_1^T \times_2 \sqrt{\boldsymbol{\Sigma}_2^{-1}}\boldsymbol{U}_2^T \times_3 \sqrt{\boldsymbol{\Sigma}_3^{-1}}\boldsymbol{U}_3^T$, then $\boldsymbol{X} = \boldsymbol{W} \times_1 \boldsymbol{H}_{:1:} \times_2 \boldsymbol{R}_{:1:} \times_3 \boldsymbol{T}_{:1:}$ is a decomposition of $\boldsymbol{X}$. Since

$$\boldsymbol{X}_{(1)} = \sqrt{\lambda_1/\lambda_4}^{\frac{-1}{\alpha}}\boldsymbol{U}_1\sqrt{\boldsymbol{\Sigma}_1}\boldsymbol{W}_{(1)}(\boldsymbol{T}_{:1:} \otimes \boldsymbol{R}_{:1:})^T = \boldsymbol{U}_1\sqrt{\boldsymbol{\Sigma}_1}\sqrt{\boldsymbol{\Sigma}_1}\boldsymbol{V}_1^T$$

$$\boldsymbol{X}_{(2)} = \sqrt{\lambda_1/\lambda_4}^{\frac{-1}{\alpha}}\boldsymbol{U}_2\sqrt{\boldsymbol{\Sigma}_2}\boldsymbol{W}_{(2)}(\boldsymbol{T}_{:1:} \otimes \boldsymbol{H}_{:1:})^T = \boldsymbol{U}_2\sqrt{\boldsymbol{\Sigma}_2}\sqrt{\boldsymbol{\Sigma}_2}\boldsymbol{V}_2^T$$

$$\boldsymbol{X}_{(3)} = \sqrt{\lambda_1/\lambda_4}^{\frac{-1}{\alpha}}\boldsymbol{U}_3\sqrt{\boldsymbol{\Sigma}_3}\boldsymbol{W}_{(3)}(\boldsymbol{R}_{:1:} \otimes \boldsymbol{H}_{:1:})^T = \boldsymbol{U}_3\sqrt{\boldsymbol{\Sigma}_3}\sqrt{\boldsymbol{\Sigma}_3}\boldsymbol{V}_3^T$$

thus

$$\boldsymbol{W}_{(1)}(\boldsymbol{T}_{:1:} \otimes \boldsymbol{R}_{:1:})^T = \sqrt{\lambda_1/\lambda_4}^{\frac{1}{\alpha}}\sqrt{\boldsymbol{\Sigma}_1}\boldsymbol{V}_1^T$$

$$\boldsymbol{W}_{(2)}(\boldsymbol{T}_{:1:} \otimes \boldsymbol{H}_{:1:})^T = \sqrt{\lambda_1/\lambda_4}^{\frac{1}{\alpha}}\sqrt{\boldsymbol{\Sigma}_2}\boldsymbol{V}_2^T$$

$$\boldsymbol{W}_{(3)}(\boldsymbol{R}_{:1:} \otimes \boldsymbol{H}_{:1:})^T = \sqrt{\lambda_1/\lambda_4}^{\frac{1}{\alpha}}\sqrt{\boldsymbol{\Sigma}_3}\boldsymbol{V}_3^T$$

If $P = D$, we have that

$$\sum_{d=1}^{D/P} \lambda_1(\|\boldsymbol{H}_{:d:}\|_F^\alpha + \|\boldsymbol{R}_{:d:}\|_F^\alpha + \|\boldsymbol{T}_{:d:}\|_F^\alpha)$$

$$+\lambda_4(\|\boldsymbol{W}_{(1)}(\boldsymbol{T}_{:d:} \otimes \boldsymbol{R}_{:d:})^T\|_F^\alpha + \|\boldsymbol{W}_{(2)}(\boldsymbol{T}_{:d:} \otimes \boldsymbol{H}_{:d:})^T\|_F^\alpha + \|\boldsymbol{W}_{(3)}(\boldsymbol{R}_{:d:} \otimes \boldsymbol{H}_{:d:})^T\|_F^\alpha)$$

$$=\lambda_1(\|\boldsymbol{H}_{:1:}\|_F^\alpha + \|\boldsymbol{R}_{:1:}\|_F^\alpha + \|\boldsymbol{T}_{:1:}\|_F^\alpha)$$

$$+\lambda_4(\|\boldsymbol{W} \times_2 \boldsymbol{R}_{:1:} \times_3 \boldsymbol{T}_{:1:}\|_F^\alpha + \|\boldsymbol{W} \times_3 \boldsymbol{T}_{:1:} \times_1 \boldsymbol{H}_{:1:}\|_F^\alpha + \|\boldsymbol{W} \times_1 \boldsymbol{H}_{:1:} \times_2 \boldsymbol{R}_{:1:}\|_F^\alpha))$$

$$=\lambda_1(\|\boldsymbol{H}_{:1:}\|_F^\alpha + \|\boldsymbol{R}_{:1:}\|_F^\alpha + \|\boldsymbol{T}_{:1:}\|_F^\alpha)$$

$$+\lambda_4(\|\boldsymbol{W}_{(1)}(\boldsymbol{T}_{:1:} \otimes \boldsymbol{R}_{:1:})^T\|_F^\alpha + \|\boldsymbol{W}_{(2)}(\boldsymbol{T}_{:1:} \otimes \boldsymbol{H}_{:1:})^T\|_F^\alpha + \|\boldsymbol{W}_{(3)}(\boldsymbol{R}_{:1:} \otimes \boldsymbol{H}_{:1:})^T\|_F^\alpha)$$

$$=\sqrt{\lambda_1\lambda_4}(\|\boldsymbol{U}_1\sqrt{\boldsymbol{\Sigma}_1}\|_F^\alpha + \|\boldsymbol{U}_2\sqrt{\boldsymbol{\Sigma}_2}\|_F^\alpha + \|\boldsymbol{U}_3\sqrt{\boldsymbol{\Sigma}_3}\|_F^\alpha)$$

$$+\sqrt{\lambda_1\lambda_4}(\|\sqrt{\boldsymbol{\Sigma}_1}\boldsymbol{V}_1^T\|_F^\alpha + \|\sqrt{\boldsymbol{\Sigma}_2}\boldsymbol{V}_2^T\|_F^\alpha + \|\sqrt{\boldsymbol{\Sigma}_3}\boldsymbol{V}_3^T\|_F^\alpha)$$

$$=\sqrt{\lambda_1\lambda_4}(\|\sqrt{\boldsymbol{\Sigma}_1}\|_F^\alpha + \|\sqrt{\boldsymbol{\Sigma}_2}\|_F^\alpha + \|\sqrt{\boldsymbol{\Sigma}_3}\|_F^\alpha) + \sqrt{\lambda_1\lambda_4}(\|\sqrt{\boldsymbol{\Sigma}_1}\|_F^\alpha + \|\sqrt{\boldsymbol{\Sigma}_2}\|_F^\alpha + \|\sqrt{\boldsymbol{\Sigma}_3}\|_F^\alpha)$$

$$=\sqrt{\lambda_1\lambda_4}(\|\boldsymbol{X}_{(1)}\|_*^\alpha + \|\boldsymbol{X}_{(2)}\|_*^\alpha + \|\boldsymbol{X}_{(3)}\|_*^\alpha + \|\boldsymbol{X}_{(1)}\|_*^\alpha + \|\boldsymbol{X}_{(2)}\|_*^\alpha + \|\boldsymbol{X}_{(3)}\|_*^\alpha)$$

$$=2L(\boldsymbol{X};\alpha)$$

$$\square$$

**Proposition 2.** *For any $\boldsymbol{X}$, and for any decomposition of $\boldsymbol{X}$, $\boldsymbol{X}$, $\boldsymbol{X} = \sum_{d=1}^{D/P} \boldsymbol{W} \times_1 \boldsymbol{H}_{:d:} \times_2 \boldsymbol{R}_{:d:} \times_3 \boldsymbol{T}_{:d:}$, we have*

$$2\sqrt{\lambda_2\lambda_3}L(\boldsymbol{X}) \leq \sum_{d=1}^{D/P} \lambda_2(\|\boldsymbol{T}_{:d:}\|_F^\alpha\|\boldsymbol{R}_{:d:}\|_F^\alpha + \|\boldsymbol{T}_{:d:}\|_F^\alpha\|\boldsymbol{H}_{:d:}\|_F^\alpha + \|\boldsymbol{R}_{:d:}\|_F^\alpha\|\boldsymbol{H}_{:d:}\|_F^\alpha)$$

$$+ \lambda_3(\|\boldsymbol{W} \times_1 \boldsymbol{H}_{:d:}\|_F^\alpha + \|\boldsymbol{W} \times_2 \boldsymbol{R}_{:d:}\|_F^\alpha + \|\boldsymbol{W} \times_3 \boldsymbol{T}_{:d:}\|_F^\alpha) \tag{8}$$

*And there exists some $\boldsymbol{X}'$, and for any decomposition of $\boldsymbol{X}'$, such that the two sides of Eq.(6) can not achieve equality.*

*Proof.* Since $\boldsymbol{X}_{(1)} = \sum_{d=1}^{D/P}(\boldsymbol{H}_{:d:}\boldsymbol{W}_{(1)})(\boldsymbol{T}_{:d:} \otimes \boldsymbol{R}_{:d:})^T$, $\boldsymbol{X}_{(2)} = \sum_{d=1}^{D/P}(\boldsymbol{R}_{:d:}\boldsymbol{W}_{(2)})(\boldsymbol{T}_{:d:} \otimes \boldsymbol{H}_{:d:})^T$, $\boldsymbol{X}_{(3)} = \sum_{d=1}^{D/P}(\boldsymbol{T}_{:d:}\boldsymbol{W}_{(3)})(\boldsymbol{R}_{:d:} \otimes \boldsymbol{H}_{:d:})^T$ and for any matrix $\boldsymbol{A}, \boldsymbol{B}$, $\|\boldsymbol{A} \otimes \boldsymbol{B}\|_F^\alpha = \|\boldsymbol{A}\|_F^\alpha\|\boldsymbol{B}\|_F^\alpha$, by applying Lemma 1 to $\boldsymbol{X}_{(1)}, \boldsymbol{X}_{(2)}, \boldsymbol{X}_{(3)}$, we have that

$$2L(\boldsymbol{X};\alpha)$$

$$\leq \sum_{d=1}^{D/P} \|(\boldsymbol{H}_{:d:}\boldsymbol{W}_{(1)})(\boldsymbol{T}_{:d:} \otimes \boldsymbol{R}_{:d:})^T\|_*^{\alpha/2} + \|(\boldsymbol{R}_{:d:}\boldsymbol{W}_{(2)})(\boldsymbol{T}_{:d:} \otimes \boldsymbol{H}_{:d:})^T\|_*^{\alpha/2} + \|(\boldsymbol{T}_{:d:}\boldsymbol{W}_{(3)})(\boldsymbol{R}_{:d:} \otimes \boldsymbol{H}_{:d:})^T\|_*^{\alpha/2}$$

$$\leq \sum_{d=1}^{D/P} \lambda(\|\boldsymbol{T}_{:d:} \otimes \boldsymbol{R}_{:d:}\|_F^\alpha + \|\boldsymbol{T}_{:d:} \otimes \boldsymbol{H}_{:d:}\|_F^\alpha + \|\boldsymbol{R}_{:d:} \otimes \boldsymbol{H}_{:d:}\|_F^\alpha)$$

$$+\frac{1}{\lambda}(\|\boldsymbol{H}_{:d:}\boldsymbol{W}_{(1)}\|_F^\alpha + \|\boldsymbol{R}_{:d:}\boldsymbol{W}_{(2)}\|_F^\alpha + \|\boldsymbol{T}_{:d:}\boldsymbol{W}_{(3)}\|_F^\alpha)$$

$$= \sum_{d=1}^{D/P} \lambda(\|\boldsymbol{T}_{:d:}\|_F^\alpha\|\boldsymbol{R}_{:d:}\|_F^\alpha + \|\boldsymbol{T}_{:d:}\|_F^\alpha\|\boldsymbol{H}_{:d:}\|_F^\alpha + \|\boldsymbol{R}_{:d:}\|_F^\alpha\|\boldsymbol{H}_{:d:}\|_F^\alpha)$$

$$+ \lambda(\|\boldsymbol{W} \times_1 \boldsymbol{H}_{:d:}\|_F^\alpha + \|\boldsymbol{W} \times_2 \boldsymbol{R}_{:d:}\|_F^\alpha + \|\boldsymbol{W} \times_3 \boldsymbol{T}_{:d:}\|_F^\alpha)$$

Let $\lambda = \sqrt{\lambda_2/\lambda_3}$, we have that

$$2\sqrt{\lambda_2\lambda_3}L(\boldsymbol{X}) \leq \sum_{d=1}^{D/P} \lambda_2(\|\boldsymbol{T}_{:d:}\|_F^\alpha\|\boldsymbol{R}_{:d:}\|_F^\alpha + \|\boldsymbol{T}_{:d:}\|_F^\alpha\|\boldsymbol{H}_{:d:}\|_F^\alpha + \|\boldsymbol{R}_{:d:}\|_F^\alpha\|\boldsymbol{H}_{:d:}\|_F^\alpha)$$

$$+ \lambda_3(\|\boldsymbol{W} \times_1 \boldsymbol{H}_{:d:}\|_F^\alpha + \|\boldsymbol{W} \times_2 \boldsymbol{R}_{:d:}\|_F^\alpha + \|\boldsymbol{W} \times_3 \boldsymbol{T}_{:d:}\|_F^\alpha) \tag{9}$$

Let $\boldsymbol{X}' \in \mathbb{R}^{2\times2\times2}$, $\boldsymbol{X}'_{1,1,1} = \boldsymbol{X}'_{2,2,2} = 1$ and $\boldsymbol{X}'_{ijk} = 0$ otherwise. Then

$$\boldsymbol{X}'_{(1)} = \boldsymbol{X}'_{(2)} = \boldsymbol{X}'_{(3)} = \left( \begin{array}{cccc} 1 & 0 & 0 & 0 \\ 0 & 0 & 0 & 1 \end{array} \right)$$

Thus $\mathrm{rank}(\boldsymbol{X}'_{(1)}) = \mathrm{rank}(\boldsymbol{X}'_{(2)}) = \mathrm{rank}(\boldsymbol{X}'_{(3)}) = 2$. In Lemma 1, a necessary condition of the equality holds is that $\mathrm{rank}(\boldsymbol{U}) = \mathrm{rank}(\boldsymbol{V}) = \mathrm{rank}(\boldsymbol{Z})$. Thus, the equality holds only if $P = D$ and

$$\mathrm{rank}(\boldsymbol{X}'_{(1)}) = \mathrm{rank}(\boldsymbol{T}_{:1:} \otimes \boldsymbol{R}_{:1:}) = 2, \mathrm{rank}(\boldsymbol{X}'_{(2)}) = \mathrm{rank}(\boldsymbol{T}_{:1:} \otimes \boldsymbol{H}_{:1:}) = 2$$

$$\mathrm{rank}(\boldsymbol{X}'_{(3)}) = \mathrm{rank}(\boldsymbol{R}_{:1:} \otimes \boldsymbol{H}_{:1:}) = 2$$

Since for any matrix $\boldsymbol{A}, \boldsymbol{B}, \mathrm{rank}(\boldsymbol{A} \otimes \boldsymbol{B}) = \mathrm{rank}(\boldsymbol{A})\,\mathrm{rank}(\boldsymbol{B})$, we have that

$$\mathrm{rank}(\boldsymbol{T}_{:1:})\,\mathrm{rank}(\boldsymbol{R}_{:1:}) = 2, \mathrm{rank}(\boldsymbol{T}_{:1:})\,\mathrm{rank}(\boldsymbol{H}_{:1:}) = 2, \mathrm{rank}(\boldsymbol{R}_{:1:})\,\mathrm{rank}(\boldsymbol{H}_{:1:}) = 2$$

The above equations have no non-negative integer solution, thus there is no decomposition of $\boldsymbol{X}'$ such that

$$2\sqrt{\lambda_2 \lambda_3} L(\boldsymbol{X}) = \sum_{d=1}^{D/P} \lambda_2 (\|\boldsymbol{T}_{:d:}\|_F^\alpha \|\boldsymbol{R}_{:d:}\|_F^\alpha + \|\boldsymbol{T}_{:d:}\|_F^\alpha \|\boldsymbol{H}_{:d:}\|_F^\alpha + \|\boldsymbol{R}_{:d:}\|_F^\alpha \|\boldsymbol{H}_{:d:}\|_F^\alpha)$$

$$+ \lambda_3 (\|\boldsymbol{W} \times_1 \boldsymbol{H}_{:d:}\|_F^\alpha + \|\boldsymbol{W} \times_2 \boldsymbol{R}_{:d:}\|_F^\alpha + \|\boldsymbol{W} \times_3 \boldsymbol{T}_{:d:}\|_F^\alpha)$$

$\square$

**Proposition 3.** *For any $\boldsymbol{X}$, and for any decomposition of $\boldsymbol{X}$, $\boldsymbol{X} = \sum_{d=1}^D \boldsymbol{W} \times_1 \boldsymbol{H}_{:d1} \times_2 \boldsymbol{R}_{:d1} \times_3 \boldsymbol{T}_{:d1}$, we have*

$$2\sqrt{\lambda_1 \lambda_4} L(\boldsymbol{X}; 2) \le 6\sqrt{\lambda_1 \lambda_4}\|\boldsymbol{X}\|_* \le \sum_{d=1}^D \lambda_1 (\|\boldsymbol{H}_{:d1}\|_F^2 + \|\boldsymbol{R}_{:d1}\|_F^2 + \|\boldsymbol{T}_{:d1}\|_F^2)$$

$$+ \lambda_4 (\|\boldsymbol{W} \times_2 \boldsymbol{R}_{:d1} \times_3 \boldsymbol{T}_{:d1}\|_F^2 + \|\boldsymbol{W} \times_3 \boldsymbol{T}_{:d1} \times_1 \boldsymbol{H}_{:d1}\|_F^2 + \|\boldsymbol{W} \times_1 \boldsymbol{H}_{:d1} \times_2 \boldsymbol{R}_{:d1}\|_F^2)$$

*where $\boldsymbol{W} = 1$.*

*Proof.* $\forall \boldsymbol{X}$, let $S_1 = \{(\boldsymbol{W}, \boldsymbol{H}, \boldsymbol{R}, \boldsymbol{T}) | \boldsymbol{X} = \sum_{d=1}^D \boldsymbol{W} \times_1 \boldsymbol{H}_{:d:} \times_2 \boldsymbol{R}_{:d:} \times_3 \boldsymbol{T}_{:d:}\}, S_2 = \{(\boldsymbol{W}, \boldsymbol{H}, \boldsymbol{R}, \boldsymbol{T}) | \boldsymbol{X} = \sum_{d=1}^D \boldsymbol{W} \times_1 \boldsymbol{H}_{:d1} \times_2 \boldsymbol{R}_{:d1} \times_3 \boldsymbol{T}_{:d1}, \boldsymbol{W} = 1\}$. We have that

$$2\sqrt{\lambda_1 \lambda_4}(\sum_{d=1}^D \|\boldsymbol{H}_{:d1}\|_F \|\boldsymbol{R}_{:d1}\|_F \|\boldsymbol{T}_{:d1}\|_F)$$

$$\le 2(\sum_{d=1}^D \lambda_4 \|\boldsymbol{R}_{:d1}\|_F^2 \|\boldsymbol{T}_{:d1}\|_F^2)^{1/2}(\sum_{d=1}^D \lambda_1 \|\boldsymbol{H}_{:d1}\|_F^2)^{1/2}$$

$$= \sum_{d=1}^D \lambda_1 \|\boldsymbol{H}_{:d1}\|_F^2 + \lambda_4 \|\boldsymbol{W} \times_2 \boldsymbol{R}_{:d:} \times_3 \boldsymbol{T}_{:d:}\|_F^2$$

where $\boldsymbol{W} = 1$. The equality holds if and only if $\lambda_4 \|\boldsymbol{R}_{:d1}\|_F^2 \|\boldsymbol{T}_{:d1}\|_F^2 = \lambda_1 \|\boldsymbol{H}_{:d1}\|_F^2$, i.e., $\sqrt{\lambda_4/\lambda_1}\|\boldsymbol{R}_{:d1}\|_2\|\boldsymbol{T}_{:d1}\|_2 = \|\boldsymbol{H}_{:d1}\|_2$. For a CP decomposition of $\boldsymbol{X}$, $\boldsymbol{X} = \sum_{d=1}^D \boldsymbol{W} \times_1 \boldsymbol{H}_{:d1} \times_2 \boldsymbol{R}_{:d1} \times_3 \boldsymbol{T}_{:d1}$, we let $\boldsymbol{H}'_{:d1} = \sqrt{\frac{\|\boldsymbol{R}_{:d1}\|_2\|\boldsymbol{T}_{:d1}\|_2}{\|\boldsymbol{H}_{:d1}\|_2}}\boldsymbol{H}_{:i}, \boldsymbol{R}'_{:i} = \sqrt{\frac{\|\boldsymbol{H}_{:d1}\|_2}{\|\boldsymbol{R}_{:d1}\|_2\|\boldsymbol{T}_{:d1}\|_2}}\boldsymbol{R}_{:i}, \boldsymbol{T}'_{:i} = \boldsymbol{T}_{:i}$ if $\|\boldsymbol{H}_{:d1}\|_2 \ne 0, \|\boldsymbol{R}_{:d1}\|_2 \ne 0, \|\boldsymbol{T}_{:d1}\|_2 \ne 0$ and otherwise $\boldsymbol{H}'_{:d} = \boldsymbol{0}, \boldsymbol{R}'_{:d} = \boldsymbol{0}, \boldsymbol{T}'_{:d} = \boldsymbol{0}$. Then $\boldsymbol{X} = \sum_{d=1}^D \boldsymbol{W} \times_1 \boldsymbol{H}'_{:d1} \times_2 \boldsymbol{R}'_{:d1} \times_3 \boldsymbol{T}'_{:d1}$ is another CP decomposition of $\boldsymbol{X}$ and $\|\boldsymbol{R}'_{:d1}\|_2\|\boldsymbol{T}'_{:d1}\|_2 = \|\boldsymbol{H}'_{:d1}\|_2$. Thus

$$2\sqrt{\lambda_1 \lambda_4}\|\boldsymbol{X}\|_* = \min_{S_2} \sum_{d=1}^D \lambda_1 \|\boldsymbol{H}_{:d1}\|_F^2 + \lambda_4 \|\boldsymbol{W} \times_2 \boldsymbol{R}_{:d:} \times_3 \boldsymbol{T}_{:d:}\|_F^2$$

Similarly, we can get

$$2\sqrt{\lambda_1 \lambda_4}\|\boldsymbol{X}\|_* = \min_{S_2} \sum_{d=1}^D \lambda_1 \|\boldsymbol{R}_{:d1}\|_F^2 + \lambda_4 \|\boldsymbol{W} \times_2 \boldsymbol{T}_{:d:} \times_3 \boldsymbol{H}_{:d:}\|_F^2$$

$$2\sqrt{\lambda_1\lambda_4}\|\boldsymbol{X}\|_* = \min_{S_2} \sum_{d=1}^{D} \lambda_1\|\boldsymbol{T}_{:d1}\|_F^2 + \lambda_4\|\boldsymbol{W} \times_2 \boldsymbol{H}_{:d:} \times_3 \boldsymbol{R}_{:d:}\|_F^2$$

By Propostion 1, we have that

$$2\sqrt{\lambda_1\lambda_4}\|\boldsymbol{X}_{(1)}\|_* = \sum_{d=1}^{D} \lambda_1\|\boldsymbol{H}_{:d1}\|_F^2 + \lambda_4\|\boldsymbol{W} \times_2 \boldsymbol{R}_{:d:} \times_3 \boldsymbol{T}_{:d:}\|_F^2$$

Since $S_2$ is a subset of $S_1$, we have that

$$\|\boldsymbol{X}_{(1)}\|_* \leq \|\boldsymbol{X}\|_*$$

Similarly, we can prove that $\|\boldsymbol{X}_{(2)}\|_* \leq \|\boldsymbol{X}\|_*$ and $\|\boldsymbol{X}_{(3)}\|_* \leq \|\boldsymbol{X}\|_*$, thus

$$2\sqrt{\lambda_1\lambda_4}L(\boldsymbol{X};2) \leq 6\sqrt{\lambda_1\lambda_4}\|\boldsymbol{X}\|_* \leq \sum_{d=1}^{D} \lambda_1(\|\boldsymbol{H}_{:d1}\|_F^2 + \|\boldsymbol{R}_{:d1}\|_F^2 + \|\boldsymbol{T}_{:d1}\|_F^2)$$
$$+ \lambda_4(\|\boldsymbol{W} \times_2 \boldsymbol{R}_{:d1} \times_3 \boldsymbol{T}_{:d1}\|_F^2 + \|\boldsymbol{W} \times_3 \boldsymbol{T}_{:d1} \times_1 \boldsymbol{H}_{:d1}\|_F^2 + \|\boldsymbol{W} \times_1 \boldsymbol{H}_{:d1} \times_2 \boldsymbol{R}_{:d1}\|_F^2)$$

$\square$

**Proposition 4.** *For any* $\boldsymbol{X}$, *there exists a decomposition of* $\boldsymbol{X} = \sum_{d=1}^{D/P} \hat{\boldsymbol{W}} \times_1 \hat{\boldsymbol{H}}_{:d:} \times_2 \hat{\boldsymbol{R}}_{:d:} \times_3 \hat{\boldsymbol{T}}_{:d:}$, *such that*

$$\sum_{d=1}^{D/P} \lambda_1(\|\hat{\boldsymbol{H}}_{:d:}\|_F^\alpha + \|\hat{\boldsymbol{R}}_{:d:}\|_F^\alpha + \|\hat{\boldsymbol{T}}_{:d:}\|_F^\alpha)$$
$$+\lambda_4(\|\hat{\boldsymbol{W}} \times_2 \hat{\boldsymbol{R}}_{:d:} \times_3 \hat{\boldsymbol{T}}_{:d:}\|_F^\alpha + \|\hat{\boldsymbol{W}} \times_3 \hat{\boldsymbol{T}}_{:d:} \times_1 \hat{\boldsymbol{H}}_{:d:}\|_F^\alpha + \|\hat{\boldsymbol{W}} \times_1 \hat{\boldsymbol{H}}_{:d:} \times_2 \hat{\boldsymbol{R}}_{:d:}\|_F^\alpha)$$
$$<\sqrt{\lambda_1\lambda_4}/\sqrt{\lambda_2\lambda_3} \sum_{d=1}^{D/P} \lambda_2(\|\hat{\boldsymbol{T}}_{:d:}\|_F^\alpha\|\hat{\boldsymbol{R}}_{:d:}\|_F^\alpha + \|\hat{\boldsymbol{T}}_{:d:}\|_F^\alpha\|\hat{\boldsymbol{H}}_{:d:}\|_F^\alpha + \|\hat{\boldsymbol{R}}_{:d:}\|_F^\alpha\|\hat{\boldsymbol{H}}_{:d:}\|_F^\alpha)$$
$$+\lambda_3(\|\hat{\boldsymbol{W}} \times_1 \hat{\boldsymbol{H}}_{:d:}\|_F^\alpha + \|\hat{\boldsymbol{W}} \times_2 \hat{\boldsymbol{R}}_{:d:}\|_F^\alpha + \|\hat{\boldsymbol{W}} \times_3 \hat{\boldsymbol{T}}_{:d:}\|_F^\alpha)$$

*Furthermore, for some* $\boldsymbol{X}$, *there exists a decomposition of* $\boldsymbol{X}$, $\boldsymbol{X} = \sum_{d=1}^{D/P} \tilde{\boldsymbol{W}} \times_1 \tilde{\boldsymbol{H}}_{:d:} \times_2 \tilde{\boldsymbol{R}}_{:d:} \times_3 \tilde{\boldsymbol{T}}_{:d:}$, *such that*

$$\sum_{d=1}^{D/P} \lambda_1(\|\tilde{\boldsymbol{H}}_{:d:}\|_F^\alpha + \|\tilde{\boldsymbol{R}}_{:d:}\|_F^\alpha + \|\tilde{\boldsymbol{T}}_{:d:}\|_F^\alpha)$$
$$+\lambda_4(\|\tilde{\boldsymbol{W}} \times_2 \tilde{\boldsymbol{R}}_{:d:} \times_3 \tilde{\boldsymbol{T}}_{:d:}\|_F^\alpha + \|\tilde{\boldsymbol{W}} \times_3 \tilde{\boldsymbol{T}}_{:d:} \times_1 \tilde{\boldsymbol{H}}_{:d:}\|_F^\alpha + \|\tilde{\boldsymbol{W}} \times_1 \tilde{\boldsymbol{H}}_{:d:} \times_2 \tilde{\boldsymbol{R}}_{:d:}\|_F^\alpha)$$
$$>\sqrt{\lambda_1\lambda_4}/\sqrt{\lambda_2\lambda_3} \sum_{d=1}^{D/P} \lambda_2(\|\tilde{\boldsymbol{T}}_{:d:}\|_F^\alpha\|\tilde{\boldsymbol{R}}_{:d:}\|_F^\alpha + \|\tilde{\boldsymbol{T}}_{:d:}\|_F^\alpha\|\tilde{\boldsymbol{H}}_{:d:}\|_F^\alpha + \|\tilde{\boldsymbol{R}}_{:d:}\|_F^\alpha\|\tilde{\boldsymbol{H}}_{:d:}\|_F^\alpha)$$
$$+\lambda_3(\|\tilde{\boldsymbol{W}} \times_1 \tilde{\boldsymbol{H}}_{:d:}\|_F^\alpha + \|\tilde{\boldsymbol{W}} \times_2 \tilde{\boldsymbol{R}}_{:d:}\|_F^\alpha + \|\tilde{\boldsymbol{W}} \times_3 \tilde{\boldsymbol{T}}_{:d:}\|_F^\alpha)$$

*Proof.* To simplify the notations, we denote $\hat{\boldsymbol{H}}_{:d:}$ as $\hat{\boldsymbol{H}}$, $\hat{\boldsymbol{R}}_{:d:}$ as $\hat{\boldsymbol{R}}$ and $\hat{\boldsymbol{T}}_{:d:}$ as $\hat{\boldsymbol{T}}$. To prove the inequality, we only need to prove

$$\sqrt{\lambda_1/\lambda_4}(\|\hat{\boldsymbol{H}}\|_F^\alpha + \|\hat{\boldsymbol{R}}\|_F^\alpha + \|\hat{\boldsymbol{T}}\|_F^\alpha)$$
$$+\sqrt{\lambda_4/\lambda_1}(\|\hat{\boldsymbol{W}} \times_2 \hat{\boldsymbol{R}} \times_3 \hat{\boldsymbol{T}}\|_F^\alpha + \|\hat{\boldsymbol{W}} \times_3 \hat{\boldsymbol{T}} \times_1 \hat{\boldsymbol{H}}\|_F^\alpha + \|\hat{\boldsymbol{W}} \times_1 \hat{\boldsymbol{H}} \times_2 \hat{\boldsymbol{R}}\|_F^\alpha)$$
$$>\sqrt{\lambda_2/\lambda_3}(\|\hat{\boldsymbol{T}}\|_F^\alpha\|\hat{\boldsymbol{R}}\|_F^\alpha + \|\hat{\boldsymbol{T}}\|_F^\alpha\|\hat{\boldsymbol{H}}\|_F^\alpha + \|\hat{\boldsymbol{R}}\|_F^\alpha\|\hat{\boldsymbol{H}}\|_F^\alpha)$$
$$+\sqrt{\lambda_3/\lambda_2}(\|\hat{\boldsymbol{W}} \times_1 \hat{\boldsymbol{H}}\|_F^\alpha + \|\hat{\boldsymbol{W}} \times_2 \hat{\boldsymbol{R}}\|_F^\alpha + \|\hat{\boldsymbol{W}} \times_3 \hat{\boldsymbol{T}}\|_F^\alpha)$$

Let $c_1 = \sqrt{\lambda_1\lambda_3}/\sqrt{\lambda_1\lambda_3}$, for any $\boldsymbol{X}$, $\boldsymbol{X} = \sum_{d=1}^{D/P} (c_1^{-3}\hat{\boldsymbol{W}}) \times_1 (c_1\hat{\boldsymbol{H}}_{:d:}) \times_2 (c_1\hat{\boldsymbol{R}}_{:d:}) \times_3 (c_1\hat{\boldsymbol{T}}_{:d:})$ is a decomposition of $\boldsymbol{X}$, thus we only need to prove

$$c_2(\|\hat{\boldsymbol{H}}\|_F^\alpha + \|\hat{\boldsymbol{R}}\|_F^\alpha + \|\hat{\boldsymbol{T}}\|_F^\alpha) + \frac{1}{c_2}(\|\hat{\boldsymbol{W}} \times_2 \hat{\boldsymbol{R}} \times_3 \hat{\boldsymbol{T}}\|_F^\alpha + \|\hat{\boldsymbol{W}} \times_3 \hat{\boldsymbol{T}} \times_1 \hat{\boldsymbol{H}}\|_F^\alpha + \|\hat{\boldsymbol{W}} \times_1 \hat{\boldsymbol{H}} \times_2 \hat{\boldsymbol{R}}\|_F^\alpha)$$
$$>c_2(\|\hat{\boldsymbol{T}}\|_F^\alpha\|\hat{\boldsymbol{R}}\|_F^\alpha + \|\hat{\boldsymbol{T}}\|_F^\alpha\|\hat{\boldsymbol{H}}\|_F^\alpha + \|\hat{\boldsymbol{R}}\|_F^\alpha\|\hat{\boldsymbol{H}}\|_F^\alpha) + \frac{1}{c_2}(\|\hat{\boldsymbol{W}} \times_1 \hat{\boldsymbol{H}}\|_F^\alpha + \|\hat{\boldsymbol{W}} \times_2 \hat{\boldsymbol{R}}\|_F^\alpha + \|\hat{\boldsymbol{W}} \times_3 \hat{\boldsymbol{T}}\|_F^\alpha)$$

where $c_2 = \sqrt{\lambda_1^2 \lambda_3}/\sqrt{\lambda_2 \lambda_4^2}$.

If $\boldsymbol{X} \in \mathbb{R}^{1 \times 1 \times 1}$, let $c = \boldsymbol{X}^2 + 100, \hat{\boldsymbol{H}} = \hat{\boldsymbol{R}} = \hat{\boldsymbol{T}} = c, \hat{\boldsymbol{W}} = \frac{\boldsymbol{X}}{c^3}$, we have

$$c_2(\|\hat{\boldsymbol{H}}\|_F^\alpha + \|\hat{\boldsymbol{R}}\|_F^\alpha + \|\hat{\boldsymbol{T}}\|_F^\alpha) + \frac{1}{c_2}(\|\hat{\boldsymbol{W}} \times_2 \hat{\boldsymbol{R}} \times_3 \hat{\boldsymbol{T}}\|_F^\alpha + \|\hat{\boldsymbol{W}} \times_3 \hat{\boldsymbol{T}} \times_1 \hat{\boldsymbol{H}}\|_F^\alpha + \|\hat{\boldsymbol{W}} \times_1 \hat{\boldsymbol{H}} \times_2 \hat{\boldsymbol{R}}\|_F^\alpha)$$

$$=c_2(\boldsymbol{X}^2 + 100)^2 + \frac{1}{c_2}\frac{\boldsymbol{X}^2}{(\boldsymbol{X}^2 + 100)^2}$$

$$<c_2(\|\hat{\boldsymbol{T}}\|_F^\alpha\|\hat{\boldsymbol{R}}\|_F^\alpha + \|\hat{\boldsymbol{T}}\|_F^\alpha\|\hat{\boldsymbol{H}}\|_F^\alpha + \|\hat{\boldsymbol{R}}\|_F^\alpha\|\hat{\boldsymbol{H}}\|_F^\alpha) + \frac{1}{c_2}(\|\hat{\boldsymbol{W}} \times_1 \hat{\boldsymbol{H}}\|_F^\alpha + \|\hat{\boldsymbol{W}} \times_2 \hat{\boldsymbol{R}}\|_F^\alpha + \|\hat{\boldsymbol{W}} \times_3 \hat{\boldsymbol{T}}\|_F^\alpha)$$

$$=c_2(\boldsymbol{X}^2 + 100)^4 + \frac{1}{c_2}\frac{\boldsymbol{X}^2}{(\boldsymbol{X}^2 + 100)^4}$$

For any $\boldsymbol{X} \in \mathbb{R}^{n_1 \times n_2 \times n_3}, \max\{n_1, n_2, n_3\} > 1$, let $\hat{\boldsymbol{W}}$ be a diagonal tensor, i.e., $\hat{\boldsymbol{W}}_{i,j,k} = 1$ if $i = j = k$ and otherwise 0 and let $\hat{\boldsymbol{W}} \in \mathbb{R}^{n_1 n_2 n_3 \times n_1 n_2 n_3}, \hat{\boldsymbol{H}} \in \mathbb{R}^{n_1 \times n_1 n_2 n_3}, \hat{\boldsymbol{R}} \in \mathbb{R}^{n_2 \times n_1 n_2 n_3}, \hat{\boldsymbol{T}} \in \mathbb{R}^{n_3 \times n_1 n_2 n_3}, \hat{\boldsymbol{H}}_{i,m} = \boldsymbol{X}_{ijk}, \hat{\boldsymbol{R}}_{j,m} = 1, \hat{\boldsymbol{T}}_{k,m} = 1, m = (i-1)n_2 n_3 + (j-1)n_3 + k$ and otherwise 0. An example of $\boldsymbol{X} \in \mathbb{R}^{2 \times 2 \times 2}$ is as follows:

$$\hat{\boldsymbol{H}} = \begin{pmatrix} \boldsymbol{X}_{1,1,1} & \boldsymbol{X}_{1,1,2} & \boldsymbol{X}_{1,2,1} & \boldsymbol{X}_{1,2,2} & 0 & 0 & 0 & 0 \\ 0 & 0 & 0 & 0 & \boldsymbol{X}_{2,1,1} & \boldsymbol{X}_{2,1,2} & \boldsymbol{X}_{2,2,1} & \boldsymbol{X}_{2,2,2} \end{pmatrix}$$

$$\hat{\boldsymbol{R}} = \begin{pmatrix} 1 & 1 & 0 & 0 & 1 & 1 & 0 & 0 \\ 0 & 0 & 1 & 1 & 0 & 0 & 1 & 1 \end{pmatrix} \hat{\boldsymbol{T}} = \begin{pmatrix} 1 & 0 & 1 & 0 & 1 & 0 & 1 & 0 \\ 0 & 1 & 0 & 1 & 0 & 1 & 0 & 1 \end{pmatrix}$$

Thus $\boldsymbol{X} = \hat{\boldsymbol{W}} \times_1 \hat{\boldsymbol{H}} \times_2 \hat{\boldsymbol{R}} \times_3 \hat{\boldsymbol{T}}$ is a decomposition of $\boldsymbol{X}$, for $\boldsymbol{X} = (c_1^{-3}\hat{\boldsymbol{W}}) \times_1 (c_1\hat{\boldsymbol{H}}) \times_2 (c_1\hat{\boldsymbol{R}}) \times_3 (c_1\hat{\boldsymbol{T}})$ we have

$$c_2(\|\hat{\boldsymbol{H}}\|_F^\alpha + \|\hat{\boldsymbol{R}}\|_F^\alpha + \|\hat{\boldsymbol{T}}\|_F^\alpha) + \frac{1}{c_2}(\|\hat{\boldsymbol{W}} \times_2 \hat{\boldsymbol{R}} \times_3 \hat{\boldsymbol{T}}\|_F^\alpha + \|\hat{\boldsymbol{W}} \times_3 \hat{\boldsymbol{T}} \times_1 \hat{\boldsymbol{H}}\|_F^\alpha + \|\hat{\boldsymbol{W}} \times_1 \hat{\boldsymbol{H}} \times_2 \hat{\boldsymbol{R}}\|_F^\alpha)$$

$$=c_2(\|\hat{\boldsymbol{H}}\|_F^\alpha + \|\hat{\boldsymbol{R}}\|_F^\alpha + \|\hat{\boldsymbol{T}}\|_F^\alpha) + \frac{1}{c_2}(\|\hat{\boldsymbol{W}}_{(1)}(\hat{\boldsymbol{T}} \otimes \hat{\boldsymbol{R}})^T\|_F^\alpha + \|\hat{\boldsymbol{W}}_{(2)}(\hat{\boldsymbol{T}} \otimes \hat{\boldsymbol{H}})^T\|_F^\alpha + \|\hat{\boldsymbol{W}}_{(3)}(\hat{\boldsymbol{R}} \otimes \hat{\boldsymbol{H}})^T\|_F^\alpha)$$

$$<\frac{1}{c_2}(\|(\hat{\boldsymbol{T}} \otimes \hat{\boldsymbol{R}})^T\|_F^\alpha + \|(\hat{\boldsymbol{T}} \otimes \hat{\boldsymbol{H}})^T\|_F^\alpha + \|(\hat{\boldsymbol{R}} \otimes \hat{\boldsymbol{H}})^T\|_F^\alpha) + c_2(\|\hat{\boldsymbol{H}}\|_F^\alpha + \|\hat{\boldsymbol{R}}\|_F^\alpha + \|\hat{\boldsymbol{T}}\|_F^\alpha)$$

$$=c_2(\|\hat{\boldsymbol{T}}\|_F^\alpha\|\hat{\boldsymbol{R}}\|_F^\alpha + \|\hat{\boldsymbol{T}}\|_F^\alpha\|\hat{\boldsymbol{H}}\|_F^\alpha + \|\hat{\boldsymbol{R}}\|_F^\alpha\|\hat{\boldsymbol{H}}\|_F^\alpha) + \frac{1}{c_2}(\|\hat{\boldsymbol{H}}\hat{\boldsymbol{W}}_{(1)}\|_F^\alpha + \|\hat{\boldsymbol{R}}\hat{\boldsymbol{W}}_{(2)}\|_F^\alpha + \|\hat{\boldsymbol{T}}\hat{\boldsymbol{W}}_{(3)}\|_F^\alpha)$$

$$=c_2(\|\hat{\boldsymbol{T}}\|_F^\alpha\|\hat{\boldsymbol{R}}\|_F^\alpha + \|\hat{\boldsymbol{T}}\|_F^\alpha\|\hat{\boldsymbol{H}}\|_F^\alpha + \|\hat{\boldsymbol{R}}\|_F^\alpha\|\hat{\boldsymbol{H}}\|_F^\alpha) + \frac{1}{c_2}(\|\hat{\boldsymbol{W}} \times_1 \hat{\boldsymbol{H}}\|_F^\alpha + \|\hat{\boldsymbol{W}} \times_2 \hat{\boldsymbol{R}}\|_F^\alpha + \|\hat{\boldsymbol{W}} \times_3 \hat{\boldsymbol{T}}\|_F^\alpha)$$

For any $\boldsymbol{X} \in \mathbb{R}^{n_1 \times n_2 \times n_3}$, let $\tilde{\boldsymbol{W}} = \boldsymbol{X}/2\sqrt{2}, \tilde{\boldsymbol{H}} = \sqrt{2}\boldsymbol{I}_{n_1 \times n_1}, \tilde{\boldsymbol{R}} = \sqrt{2}\boldsymbol{I}_{n_2 \times n_2}, \tilde{\boldsymbol{T}} = \sqrt{2}\boldsymbol{I}_{n_3 \times n_3}$, thus

$$c_2(\|\tilde{\boldsymbol{H}}\|_F^\alpha + \|\tilde{\boldsymbol{R}}\|_F^\alpha + \|\tilde{\boldsymbol{T}}\|_F^\alpha) + \frac{1}{c_2}(\|\tilde{\boldsymbol{W}} \times_2 \tilde{\boldsymbol{R}} \times_3 \tilde{\boldsymbol{T}}\|_F^\alpha + \|\tilde{\boldsymbol{W}} \times_3 \tilde{\boldsymbol{T}} \times_1 \tilde{\boldsymbol{H}}\|_F^\alpha + \|\tilde{\boldsymbol{W}} \times_1 \tilde{\boldsymbol{H}} \times_2 \tilde{\boldsymbol{R}}\|_F^\alpha)$$

$$=c_2(2n_1 + 2n_2 + 2n_3) + \frac{1}{c_2}(\|\boldsymbol{X}_{(1)}\|_F^2/2 + \|\boldsymbol{X}_{(2)}\|_F^2/2 + \|\boldsymbol{X}_{(3)}\|_F^2/2)$$

$$c_2(\|\tilde{\boldsymbol{T}}\|_F^\alpha\|\tilde{\boldsymbol{R}}\|_F^\alpha + \|\tilde{\boldsymbol{T}}\|_F^\alpha\|\tilde{\boldsymbol{H}}\|_F^\alpha + \|\tilde{\boldsymbol{R}}\|_F^\alpha\|\tilde{\boldsymbol{H}}\|_F^\alpha) + \frac{1}{c_2}(\|\tilde{\boldsymbol{W}} \times_1 \tilde{\boldsymbol{H}}\|_F^\alpha + \|\tilde{\boldsymbol{W}} \times_2 \tilde{\boldsymbol{R}}\|_F^\alpha + \|\tilde{\boldsymbol{W}} \times_3 \tilde{\boldsymbol{T}}\|_F^\alpha)$$

$$=c_2(4n_3 n_2 + 4n_3 n_1 + 4n_2 n_1) + \frac{1}{c_2}(\|\boldsymbol{X}_{(1)}\|_F^2/4 + \|\boldsymbol{X}_{(2)}\|_F^2/4 + \|\boldsymbol{X}_{(3)}\|_F^2/4)$$

Thus if $\|\boldsymbol{X}_{(1)}\|_F^2 > 16c_2^2(n_3 n_2 + n_3 n_1 + n_2 n_1)$, we have

$$c_2(\|\tilde{\boldsymbol{H}}\|_F^\alpha + \|\tilde{\boldsymbol{R}}\|_F^\alpha + \|\tilde{\boldsymbol{T}}\|_F^\alpha) + \frac{1}{c_2}(\|\tilde{\boldsymbol{W}} \times_2 \tilde{\boldsymbol{R}} \times_3 \tilde{\boldsymbol{T}}\|_F^\alpha + \|\tilde{\boldsymbol{W}} \times_3 \tilde{\boldsymbol{T}} \times_1 \tilde{\boldsymbol{H}}\|_F^\alpha + \|\tilde{\boldsymbol{W}} \times_1 \tilde{\boldsymbol{H}} \times_2 \tilde{\boldsymbol{R}}\|_F^\alpha)$$

$$>c_2(\|\tilde{\boldsymbol{T}}\|_F^\alpha\|\tilde{\boldsymbol{R}}\|_F^\alpha + \|\tilde{\boldsymbol{T}}\|_F^\alpha\|\tilde{\boldsymbol{H}}\|_F^\alpha + \|\tilde{\boldsymbol{R}}\|_F^\alpha\|\tilde{\boldsymbol{H}}\|_F^\alpha) + \frac{1}{c_2}(\|\tilde{\boldsymbol{W}} \times_1 \tilde{\boldsymbol{H}}\|_F^\alpha + \|\tilde{\boldsymbol{W}} \times_2 \tilde{\boldsymbol{R}}\|_F^\alpha + \|\tilde{\boldsymbol{W}} \times_3 \tilde{\boldsymbol{T}}\|_F^\alpha)$$

end the proof. $\qquad\qquad\square$

## C   Experimental details

**Datasets**   The statistics of the datasets, WN18RR, FB15k-237, YAGO3-10 and Kinship, are shown in Table 4.

Table 4: The statistics of the datasets.

| Dataset | #entity | #relation | #train | #valid | #test |
|---|---|---|---|---|---|
| WN18RR | 40,943 | 11 | 86,835 | 3,034 | 3,134 |
| FB15k-237 | 14,541 | 237 | 272,115 | 17,535 | 20,466 |
| YGAO3-10 | 123,188 | 37 | 1,079,040 | 5,000 | 5,000 |
| Kinship | 104 | 25 | 8,548 | 1,069 | 1,069 |

Table 5: The settings for total embedding dimension $D$ and number of parts $P$.

| | WN18RR | | FB15k-237 | | YAGO3-10 | |
|---|---|---|---|---|---|---|
| | $D$ | $P$ | $D$ | $P$ | $D$ | $P$ |
| CP | 2000 | 1 | 2000 | 1 | 1000 | 1 |
| ComplEx | 4000 | 2 | 4000 | 2 | 2000 | 2 |
| SimplE | 4000 | 2 | 4000 | 2 | 2000 | 2 |
| ANALOGY | 4000 | 4 | 4000 | 4 | 2000 | 4 |
| QuatE | 4000 | 4 | 4000 | 4 | 2000 | 4 |
| TuckER | 256 | 1 | 256 | 1 | 256 | 1 |

Table 6: The settings for power $\alpha$ and regularization coefficients $\lambda_i$.

| | WN18RR | | | FB15k-237 | | | YAGO3-10 | | |
|---|---|---|---|---|---|---|---|---|---|
| | $\alpha$ | $\lambda_1$ | $\lambda_2$ | $\alpha$ | $\lambda_1$ | $\lambda_2$ | $\alpha$ | $\lambda_1$ | $\lambda_2$ |
| CP | 3.0 | 05 | 0.07 | 2.25 | 0.005 | 0.07 | 2.25 | 0.0007 | 0.005 |
| ComplEx | 3.0 | 0.05 | 0.03 | 2.25 | 0.005 | 0.07 | 2.25 | 0.0005 | 0.007 |
| SimplE | 3.0 | 0.03 | 0.07 | 2.25 | 0.005 | 0.07 | 2.25 | 0.0007 | 0.007 |
| ANALOGY | 3.0 | 0.01 | 0.07 | 2.25 | 0.005 | 0.07 | 2.25 | 0.0007 | 0.007 |
| QuatE | 3.0 | 0.07 | 0.03 | 2.25 | 0.005 | 0.05 | 2.25 | 0.0005 | 0.005 |
| TuckER | 2.0 | 0.01 | 0.03 | 2.0 | 0.001 | 0.01 | 2.0 | 0.0003 | 0.003 |

**Evaluation Metrics**   MR$=\frac{1}{N}\sum_{i=1}^{N}\mathrm{rank}_i$, where $\mathrm{rank}_i$ is the rank of $i$th triplet in the test set and $N$ is the number of the triplets. Lower MR indicates better performance.

MRR$=\frac{1}{N}\sum_{i=1}^{N}\frac{1}{\mathrm{rank}_i}$. Higher MRR indicates better performance.

Hits@N $= \frac{1}{N}\sum_{i=1}^{N}\mathbb{I}(\mathrm{rank}_i \leq N)$, where $\mathbb{I}(\cdot)$ is the indicator function. Hits@N is the ratio of the ranks that no more than $N$, Higher Hits@N indicates better performance.

**Hyper-parameters**   We use a heuristic approach to choose the hyper-parameters and reduce the computation cost with the help of Hyperopt, a hyper-parameter optimization framework based on TPE [Bergstra et al., 2011].

For the hyper-parameter $\alpha$, we search for the best $\alpha$ in $\{2.0, 2.25, 2.5, 2.75, 3.0, 3.25, 3.5\}$ with fixed hyper-parameters $\lambda_i = 0.0001$.

For the hyper-parameter $\lambda_i$, we set $\lambda_1 = \lambda_3$ and $\lambda_2 = \lambda_4$ for all models to reduce the number of hyper-parameters because we notice that the first row of Eq.(4) $\|\boldsymbol{H}_{id:}\|_F^{\alpha} + \|\boldsymbol{R}_{jd:}\|_F^{\alpha} + \|\boldsymbol{T}_{kd:}\|_F^{\alpha}$ is equal to the third row of Eq.(4) $\|\boldsymbol{W} \times_1 \boldsymbol{H}_{id:}\|_F^{\alpha} + \|\boldsymbol{W} \times_2 \boldsymbol{R}_{jd:}\|_F^{\alpha} + \|\boldsymbol{W} \times_3 \boldsymbol{T}_{kd:}\|_F^{\alpha}$ and the second row of Eq.(4) $\|\boldsymbol{T}_{kd:}\|_F^{\alpha}\|\boldsymbol{R}_{jd:}\|_F^{\alpha} + \|\boldsymbol{T}_{kd:}\|_F^{\alpha}\|\boldsymbol{H}_{id:}\|_F^{\alpha} + \|\boldsymbol{R}_{jd:}\|_F^{\alpha}\|\boldsymbol{H}_{id:}\|_F^{\alpha}$ is equal to the

Table 7: The results on WN18RR dataset with different $\alpha$.

| $\alpha$ | 2.0 | 2.25 | 2.5 | 2.75 | 3.0 | 3.25 | 3.5 |
|---|---|---|---|---|---|---|---|
| MRR | 0.483 | 0.486 | 0.485 | 0.486 | 0.494 | 0.486 | 0.487 |
| H@1 | 0.443 | 0.445 | 0.441 | 0.442 | 0.449 | 0.443 | 0.444 |
| H@10 | 0.556 | 0.564 | 0.573 | 0.572 | 0.581 | 0.572 | 0.570 |

fourth row of Eq.(4) $\|\boldsymbol{W} \times_2 \boldsymbol{R}_{jd:} \times_3 \boldsymbol{T}_{kd:}\|_F^\alpha + \|\boldsymbol{W} \times_3 \boldsymbol{T}_{kd:} \times_1 \boldsymbol{H}_{id:}\|_F^\alpha + \|\boldsymbol{W} \times_1 \boldsymbol{H}_{id:} \times_2 \boldsymbol{R}_{jd:}\|_F^\alpha$ for CP and ComplEx. We then search the regularization coefficients $\lambda_1$ and $\lambda_2$ in $\{0.001, 0.003, 0.005, 0.007, 0.01, 0.03, 0.05, 0.07\}$ for WN18RR dataset and FB15k-237 dataset, and search $\lambda_1$ and $\lambda_2$ in $\{0.0001, 0.0003, 0.0005, 0.0007, 0.001, 0.003, 0.05, 0.007\}$ for YAGO3-10 dataset. Thus, we need 64 runs for each model to find the best $\lambda_i$. To further reduce the number of runs, we use Hyperopt, a hyper-parameter optimization framework based on TPE [Bergstra et al., 2011], to tune hyper-parameters. In our experiments, we only need 20 runs to find the best $\lambda_i$.

We use Adagrad [Duchi et al., 2011] with learning rate 0.1 as the optimizer. We set the batch size to 100 for WN18RR dataset and FB15k-237 dataset and 1000 for YAGO3-10 dataset. We train the models for 200 epochs. The settings for total embedding dimension $D$ and number of parts $P$ are shown in Table 5. The settings for power $\alpha$ and regularization coefficients $\lambda_i$ are shown in Table 6.

**Random Initialization**    We run each model three times with different random seeds and report the mean results. We do not report the error bars because our model has very small errors with respect to random initialization. The standard deviations of the results are very small. For example, the standard deviations of MRR, H@1 and H@10 of CP model with our regularization are 0.00037784, 0.00084755 and 0.00058739 on WN18RR dataset, respectively. This indicates that our model is not sensitive to the random initialization.

**The hyper-parameter** $\alpha$    We analyze the impact of the hyper-parameter, the power of the Frobenius norm $\alpha$. We run experiments on WN18RR dataset with ComplEx model. We set $\alpha$ to $\{2.0, 2.25, 2.5, 2.75, 3.0, 3.25, 3.5\}$. See Table 7 for the results.

The results show that the performance generally increases as $\alpha$ increases and then decreases as $\alpha$ increases. The best $\alpha$ for WN18RR dataset is 3.0. Therefore, we should set a more appropriate $\alpha$ value to obtain better performance.

**The hyper-parameter** $\lambda_i$    We analyze the impact of the hyper-parameter, the regularization coefficient $\lambda_i$. We run experiments on WN18RR dataset with ComplEx model. We set $\lambda_i$ to $\{0.001, 0.003, 0.005, 0.007, 0.01, 0.03, 0.05, 0.07, 0.1\}$. See Table 8 and Table 9 for the results.

The experimental results show that the model performance first increases and then decreases with the increase of $\lambda_i$, without any oscillation. Thus, we can choose suitable regularization coefficients to prevent overfitting while maintaining the expressiveness of TDB models as much as possible.

**The Number of Parts** $P$    In Section 3.1, we show that the number of parts $P$ can affect the expressiveness and computation. Thus, we study the impact of $P$ on the model performance. We evaluate the model on WN18RR dataset. We set the total embedding dimension $D$ to 256, and set the parts $P$ to $\{1, 2, 4, 8, 16, 32, 64, 128, 256\}$. See Table 10 for the results. The time is the AMD Ryzen 7 4800U CPU running time on the test set.

The results show that the model performance generally improves and the running time generally increases as $P$ increases. Thus, the larger the part $P$, the more expressive the model and the more the computation.

**A Pseudocode for IVR**    We present a pseudocode of our method in Alg.(1).

Table 8: The performance of ComplEx on WN18RR dataset with different $\lambda_1$.

| $\lambda_1$ | 0.001 | 0.003 | 0.005 | 0.007 | 0.01 | 0.03 | 0.05 | 0.07 | 0.1 |
|---|---|---|---|---|---|---|---|---|---|
| MRR | 0.473 | 0.475 | 0.476 | 0.476 | 0.480 | 0.487 | 0.491 | 0.486 | 0.467 |
| H@1 | 0.435 | 0.436 | 0.436 | 0.436 | 0.439 | 0.442 | 0.445 | 0.436 | 0.411 |
| H@10 | 0.545 | 0.555 | 0.555 | 0.557 | 0.562 | 0.576 | 0.583 | 0.585 | 0.570 |

Table 9: The performance of ComplEx on WN18RR dataset with different $\lambda_2$.

| $\lambda_2$ | 0.001 | 0.003 | 0.005 | 0.007 | 0.01 | 0.03 | 0.05 | 0.07 | 0.1 |
|---|---|---|---|---|---|---|---|---|---|
| MRR | 0.464 | 0.464 | 0.464 | 0.465 | 0.465 | 0.467 | 0.471 | 0.473 | 0.472 |
| H@1 | 0.429 | 0.430 | 0.430 | 0.430 | 0.432 | 0.432 | 0.435 | 0.437 | 0.435 |
| H@10 | 0.528 | 0.528 | 0.529 | 0.529 | 0.530 | 0.536 | 0.540 | 0.540 | 0.540 |

Table 10: The results on WN18RR dataset with different $P$.

| $P$ | 1 | 2 | 4 | 8 | 16 | 32 | 64 | 128 | 256 |
|---|---|---|---|---|---|---|---|---|---|
| MRR | 0.437 | 0.449 | 0.455 | 0.455 | 0.463 | 0.466 | 0.485 | 0.497 | 0.501 |
| H@1 | 0.405 | 0.413 | 0.413 | 0.415 | 0.425 | 0.428 | 0.446 | 0.456 | 0.460 |
| H@10 | 0.499 | 0.520 | 0.536 | 0.533 | 0.536 | 0.539 | 0.565 | 0.574 | 0.579 |
| Time | 1.971s | 2.096s | 2.111s | 2.145s | 2.301s | 2.704s | 3.520s | 6.470s | 16.336s |

---

**Algorithm 1** A pseudocode for IVR

---

**Input:**
    Core tensor: $\boldsymbol{W}$, embeddings: $(\boldsymbol{H}, \boldsymbol{R}, \boldsymbol{T})$, triplet: $(i, j, k)$.
    Regularization coefficients: $\lambda_l(l = 1, 2, 3, 4)$, the number of parts $P$. power coefficients: $\alpha$
**Output:**
    Output of model Eq.(1): $\boldsymbol{X}_{ijk}$, IVR regularization Eq.(6): $\text{reg}(\boldsymbol{X}_{ijk})$
1: Initialization: $\text{reg} := 0, \boldsymbol{x}_{1d} := \boldsymbol{H}_{id:}, \boldsymbol{x}_{2d} := \boldsymbol{R}_{jd:}, \boldsymbol{x}_{3d} := \boldsymbol{T}_{kd:}$
2: $\text{reg} := \text{reg} + \sum_{d=1}^{D/P} \lambda_1(\|\boldsymbol{x}_{1d}\|_F^\alpha + \|\boldsymbol{x}_{2d}\|_F^\alpha + \|\boldsymbol{x}_{3d}\|_F^\alpha)$
3: $\text{reg} := \text{reg} + \sum_{d=1}^{D/P} \lambda_2(\|\boldsymbol{x}_{1d}\|_F^\alpha\|\boldsymbol{x}_{2d}\|_F^\alpha + \|\boldsymbol{x}_{1d}\|_F^\alpha\|\boldsymbol{x}_{3d}\|_F^\alpha + \|\boldsymbol{x}_{2d}\|_F^\alpha\|\boldsymbol{x}_{3d}\|_F^\alpha)$
4: $\boldsymbol{x}_{1d} := \boldsymbol{W} \times_1 \boldsymbol{H}_{id:}, \boldsymbol{x}_{2d} := \boldsymbol{W} \times_2 \boldsymbol{R}_{jd:}, \boldsymbol{x}_{3d} := \boldsymbol{W} \times_3 \boldsymbol{T}_{kd:}$
5: $\text{reg} := \text{reg} + \sum_{d=1}^{D/P} \lambda_3(\|\boldsymbol{x}_{1d}\|_F^\alpha + \|\boldsymbol{x}_{2d}\|_F^\alpha + \|\boldsymbol{x}_{3d}\|_F^\alpha)$
6: $\boldsymbol{x}_{1d} := \boldsymbol{x}_{1d} \times_2 \boldsymbol{R}_{jd:}, \boldsymbol{x}_{2d} := \boldsymbol{x}_2 \times_3 \boldsymbol{T}_{kd:}, \boldsymbol{x}_{3d} := \boldsymbol{x}_{3d} \times_1 \boldsymbol{H}_{id:}$
7: $\text{reg} := \text{reg} + \sum_{d=1}^{D/P} \lambda_4(\|\boldsymbol{x}_{1d}\|_F^\alpha + \|\boldsymbol{x}_{2d}\|_F^\alpha + \|\boldsymbol{x}_{3d}\|_F^\alpha)$
8: $\boldsymbol{x} := \sum_{d=1}^{D/P} \boldsymbol{x}_{1d} \times_3 \boldsymbol{T}_{kd:}$
9: **return:** $\boldsymbol{x}, \text{reg}$

---

