# OpenReview forum: "Knowledge Graph Completion by Intermediate Variables Regularization"
_NeurIPS.cc/2024/Conference — NeurIPS 2024 poster_

### Official Review · Reviewer_YvVe · 2024-07-01

**Soundness:** 3
**Presentation:** 2
**Contribution:** 2
**Rating:** 5
**Confidence:** 3

**Summary:**

This paper proposes a general model for regularizing a variety of tensor-decomposition-based knowledge graph completion models (based on variants of real / complex CP decomposition, Tucker decomposition, and others). The authors first observe that all of these tensor decomposition models can be expressed as a block-term decomposition of a general dense tensor, with various restrictions placed on the block term core tensors / factor matrices. The authors then use their general tensor model structure to determine necessary (but not sufficient) rank-based conditions on whether a particular knowledge graph model can express symmetry, antisymmetry, and inverse relations. The authors then propose a regularization strategy for this general tensor decomposition model based on the norms of the factor matrices and various intermediate terms that arise when contracting the tensor diagram corresponding to the block term decomposition. The authors evaluate their regularization strategy on three well-known knowledge graph datasets and six different knowledge graph completion models, showing accuracy improvements resulting from their regularization strategy.

**Strengths:**

The rank-based justification of whether particular knowledge graph models can learn certain logical rules is interesting, since it explores these models from a linear algebraic perspective. The use of the hyperparameter tuner was a good idea as well, given that prior studies have established that careful hyperparameter tuning specifically for knowledge graph completion can significantly boost model performance (https://openreview.net/forum?id=BkxSmlBFvr). The knowledge graph regularization strategy appears to have non-negligible effects on accuracy, and the results are competitive with other state-of-the-art papers reporting results on the FB15K-237, YAGO, and WN18RR datasets. This experiments in this paper seem to suggest that many state of the art knowledge graph models can benefit (in significant ways) from regularization beyond what the literature currently suggests.

**Weaknesses:**

The utility of Theorem 1 is somewhat unclear to me, given that the proofs of the symmetry / anti-symmetry / inverse properties for individual models such as CP, ComplEX, etc. are sufficiently simple. For example, the fact that that CP decomposition can learn the symmetry rules follows from a single step from the commutativity of the three-vector dot product mentioned in this paper if any pair of arguments is exchanged. The symmetry / anti-symmetry proofs for ComplEX are similar. As the authors point out, the usefulness of Theorem 1 stems from its ability to give a unified proof and to prove whether other, more complex tensor decomposition models could learn such logical rules. Again, in this case, it’s not clear whether applying this rank-based theorem would be simpler than specialized proofs, or would provide any more insight into these models. Although this theorem is interesting by bringing a linear algebraic perspective to the problem, I would like to see a case where this theorem proves a property / gives a result that cannot be derived more simply through other means.

I have a second question about Theorem 1 as it relates to DistMult and TuckER; see below.

The proposed regularizer has theoretical justification, although it adds a significant number of terms to the loss function. I'm somewhat skeptical about the novelty of the regularization method that arises from tacking on the variety of intermediate terms (while leaving out others), but I also acknowledge that the experiments indicate indicate that there is room for additional progress in knowledge graph model regularization.

The authors note that this does not increase the asymptotic complexity, but the number of terms added gives me some pause. I would like to see some data about the additional runtime, if any, required in practice by adding these regularization terms. The proposed regularizer appears to improve accuracy in most case, although at what cost in additional computational runtime in practice?

Overall, the experiments in this paper are thorough. I am skeptical about the utility of Theorem 1 and the utility of viewing these models through the lens of a block-term decomposition. The experiments indicate that adding several regularization terms improves the accuracy of tensor completion models in general; this has some novelty in that it shows that knowledge graph models are heavily overparameterized. I would also like to see the runtime impact of adding several non-trivial terms to the loss function, even if the asymptotic complexity does not change.

**Questions:**

1. There is an application of Theorem 1 for ComplEX (P=2), and I was able to follow both the proof of the theorem and its application. Can the authors also provide the corresponding analysis for ANALOGY and QuatE? I would like to see this theorem applied for cases P > 2 (and even an example for the case P = 1 would be helpful. I feel as if this theorem becomes more difficult to apply and less practical as the permutation matrix becomes larger. For P = 1, I believe the condition in Theorem 1 is exactly that W is a symmetric tensor.

2. As the authors point out, DistMult forces H = T to enable the model to learn symmetric relations. TuckER is similar. Does Theorem 1 account for such restrictions? It would seem that Theorem 1 in its current form puts no conditions on H and T.

3. Line 197: When you refer to the “asymptotic complexity of Equation 4, I presume this means the cost to compute the expression. Does the cost of computing the derivative of the expression with respect to a single (h, r, t) tuple also remain the same?

---

> ### Author Rebuttal · Authors · 2024-08-06
>
> We appreciate your careful and constructive comments. We have addressed the questions that you raised as follows. Please let us know if you have any further concerns.
>
> $\textbf{Q1:}$ It’s not clear whether applying this rank-based theorem would be simpler than specialized proofs, or would provide any more insight into these models. Although this theorem is interesting by bringing a linear algebraic perspective to the problem, I would like to see a case where this theorem proves a property/gives a result that cannot be derived more simply through other means.
>
> $\textbf{A1:}$ Theorem 1 transforms the problem of learning logical rules in the TDB model into a linear algebraic problem. By Theorem 1, we only need to calculate the ranks of several matrices. We can calculate ranks through software such as PyTorch and Matlab without manually calculating them.
>
> Previous proofs of the problem of learning logical rules involved complex algebraic expressions computations, such as those in the paper of QuatE [1], particularly challenging for large $P$. In contrast, matrix operations are generally simpler and can be performed in parallel, making the proofs utilizing Theorem 1 more straightforward and easier than previous proofs, especially for large $P$.
>
> One potential application of Theorem 1 is to use these conditions as constraints in training to ensure that TDB models are capable of learning logical rules.
>
>  [1] Shuai Zhang, Yi Tay, Lina Yao, and Qi Liu. Quaternion knowledge graph embeddings. In Proceedings of the 33rd International Conference on Neural Information Processing Systems, pages 2735–2745, 2019.
>
> $\textbf{Q2:}$ Can the authors also provide the corresponding analysis of Theorem 1 for ANALOGY and QuatE? I would like to see this theorem applied for cases $P>2$. For $P=1$, I believe the condition in Theorem 1 is exactly that $\mathbf{W}$ is a symmetric tensor. I feel as if this theorem becomes more difficult to apply and less practical as the permutation matrix becomes larger.
>
> $\textbf{A2:}$
> For $P=1$, $\mathbf{W}\in \mathbb{R}^{1\times 1\times 1}$ is a scalar, then we have that $\text{rank}(\mathbf{W}\_{(2)}^{T}-\mathbf{S}\mathbf{W}\_{(2)}^{T})=0<1=P$, $\text{rank}(\mathbf{W}\_{(2)}^{T}+\mathbf{S}\mathbf{W}\_{(2)}^{T})=1=P$, $\text{rank}(\mathbf{W}\_{(2)}^{T})=\text{rank}([\mathbf{W}\_{(2)},\mathbf{S}{W}\_{(2)}^{T}])=1$, thus TDB model with $P=1$ is able to learn the symmetry rules and inverse rules.
>
> For QuatE with $P=4$, $\mathbf{W}\in \mathbb{R}^{4\times 4\times 4}$, then we have that $\text{rank}(\mathbf{W}\_{(2)}^{T}-\mathbf{S}\mathbf{W}\_{(2)}^{T})=3<4=P$, $\text{rank}(\mathbf{W}\_{(2)}^{T}+\mathbf{S}\mathbf{W}\_{(2)}^{T})=1<4=P$, $\text{rank}(\mathbf{W}\_{(2)}^{T})=\text{rank}([\mathbf{W}\_{(2)},\mathbf{S}\mathbf{W}\_{(2)}^{T}])=4$, thus, QuatE is able to learn the symmetry rules, antisymmetry rules and inverse rules. ANALOGY is similar.
>
> The permutation operation in Theorem 1 can be efficiently computed by permuating the dimensions of $\mathbf{W}$. For example, we can use the “permuate()” function in PyTorch to compute it.
>
> $\textbf{Q3:}$ As the authors point out, DistMult forces $\mathbf{H} = \mathbf{T}$ to enable the model to learn symmetric relations. TuckER is similar. Does Theorem 1 account for such restrictions? It would seem that Theorem 1 in its current form puts no conditions on $\mathbf{H}$ and $\mathbf{T}$.
>
> $\textbf{A3:}$ Yes. Theorem1 involes the constraint $\mathbf{H}=\mathbf{T}$, which mainly aims to reduce overfitting [2]. Existing TDB models for KGC except CP all forces $\mathbf{H}=\mathbf{T}$. The proof of Theorem 1 has used this constraint. We may omit some details, leading to such misunderstandings. We add the following steps to make the proof complete.
>
> According to the symmetry rules, for any triplet $(i,j,k)$, we have that $f(i,j,k)=f(k,j,i)$, i.e., $f(\mathbf{H}\_{i:}, \mathbf{R}\_{j:}, \mathbf{T}\_{k:})= f(\mathbf{H}\_{k:}, \mathbf{R}\_{j:}, \mathbf{T}\_{i:})= f(\mathbf{T}\_{k:}, \mathbf{R}\_{j:}, \mathbf{H}\_{i:})$ (we use $\mathbf{H}=\mathbf{T}$ here). We replace $\mathbf{H}\_{i:}$ by $\mathbf{h}$, replace $\mathbf{R}\_{j:}$ by $\mathbf{r}$, replace $\mathbf{T}\_{k:}$ by $\mathbf{t}$, then we have $f(\mathbf{h},\mathbf{r},\mathbf{t})=f(\mathbf{t},\mathbf{r},\mathbf{h})$, i.e., the first row of the proof of Theorem 1. We will make the proof clearer in the revision.
>
> For CP model, which does not involve the constraint, we can easily proof that it is able to learn the symmetry rules, antisymmetry rules and inverse rules.
>
> [2] Kadlec, R., Bajgar, O., and Kleindienst, J. Knowledge Base Completion: Baselines Strike Back. In Proceedings of the 2nd Workshop on Representation Learning for NLP, pp. 69–74, 2017.
>
> $\textbf{Q4:}$ I would like to see some data about the additional runtime, if any, required in practice by adding these regularization terms. The proposed regularizer appears to improve accuracy in most case, although at what cost in additional computational runtime in practice?
>
> $\textbf{A4:}$ The regularization terms only increase the training time. The training time per epoch and MRR metric are shown in the following table. IVR slightly increases the running time but enhances performance. We also provide the running time regarding hyperparameter $P$ on Page 22.
>
> Table: The running time and MRR metric of ComplEx model on WN18RR dataset.
> |Model|Time|MRR|
> |-|-|-|
> |ComplEx|36s|0.464|
> |ComplEx-F2|39s|0.467|
> |ComplEx-N3|39s|0.491|
> |ComplEx-DURA|41s|0.484|
> |ComplEx-IVR|44s|0.494|
>
> $\textbf{Q5:}$ Line 197: When you refer to the “asymptotic complexity of Equation 4, I presume this means the cost to compute the expression. Does the cost of computing the derivative of the expression with respect to a single $(h,r,t)$ tuple also remain the same?
>
> $\textbf{A5:}$ Yes. Since $\frac{\partial ||\mathbf{A}||\_F^{\alpha}}{\partial \mathbf{A}}=\alpha ||\mathbf{A}||\_F^{\alpha-2}\mathbf{A}$, the cost of computing the derivative remains the same.

---

> ### Author Response · Authors · 2024-08-13
> **The final day for discussions**
>
> Dear reviewer:
>
> Thank you once again for your meticulous comments. As today is the final day for discussions, we kindly request your feedback once more. We anticipate that our brief responses will not require much of your time. Your feedback is invaluable to us, and we eagerly await your response.

---

> > ### Comment · Reviewer_YvVe · 2024-08-14
> > **Response Acknowledged**
> >
> > I thank the authors for their response. I'm increasing my score to a 5, on the basis of the following: it does seem that even strong knowledge graph models can be improved by regularization, and the rank based analysis is intriguing. I echo the other reviewer's assessment that this framework can be viewed through the lens of a block term decomposition.

---

### Official Review · Reviewer_gAVe · 2024-07-12

**Soundness:** 3
**Presentation:** 3
**Contribution:** 3
**Rating:** 6
**Confidence:** 4

**Summary:**

This paper proposes a general framework for tensor decomposition methods on knowledge graph completion. Based on the proposed framework, the authors further introduce a novel regularization method that regularizes the norms of intermediate variables in tensor decomposition. Theoretical analysis demonstrates that regularization on intermediate variables provably upper bounds the original loss, and empirical results on different data sets verify the theoretical analysis and show significant improvements of the proposed method when combined with various knowledge graph completion methods.

**Strengths:**

- The proposed method is clearly introduced and easy to understand
- Theoretical analysis is sound and well supports the proposed method
- Empirical results on different data sets demonstrate that the proposed method improves upon existing KG completion methods by tensor decomposition models

**Weaknesses:**

Several detail points about the proposed method are not clear enough

**Questions:**

- Despite the Frobenius norm, can IVR also use other matrix norms (e.g., the spectral norm, though the computation cost may be a problem)? Some additional experiments may be useful to better understand how the proposed regularization affect model training.
- The captions of table 8 and 9 seem to contain an error. Should these tables be for different data sets?

**Limitations:**

This paper does not have direct potential negative societal impact from my perspective.

---

> ### Author Rebuttal · Authors · 2024-08-06
>
> We appreciate your careful and constructive comments. We have addressed the questions that you raised as follows. Please let us know if you have any further concerns.
>
> $\textbf{Q1:}$ Several detail points about the proposed method are not clear enough.
>
> $\textbf{A1:}$ We will make our statements clearer in the revision.
>
> $\textbf{Q2:}$ Despite the Frobenius norm, can IVR also use other matrix norms (e.g., the spectral norm, though the computation cost may be a problem)? Some additional experiments may be useful to better understand how the proposed regularization affect model training.
>
> $\textbf{A2:}$ We choose the Frobenius norm because it can be computed efficiently and is conducive to theoretical analysis. In Section 3.3, we leverage the relationship between the Frobenius norm and the trace norm to demonstrate that IVR serves as an upper bound for the overlapped trace norm of the predicted tensor.
>
> The spectral norm, with its computational complexity of $\mathcal{O}(n^3)$, poses challenges for practical computation. Establishing a theoretical framework for alternative norms may also be difficult.
>
> We have conducted several experiments to show how IVR affect model training.
>
> First, in Section 4.4, we verify that minimizing the upper bounds IVR can effectively minimize the overlapped trace norm. Lower values of overlapped trace norm encourage higher correlations among entities and relations, and thus bring a strong constraint for training.
>
> Second, we analyze the impact of hyper-parameters on model performance in Appendix C. You can refer to the Paragraph "The hyper-parameter $\alpha$", "The hyper-parameter $\lambda_{i}$" and “The Number of Parts $P$” on Page 22.
>
> Furthermore, we add an experiment to demonstrate the effect of IVR on training time. The regularization terms only increase the training time. The training time per epoch and MRR metric are shown in the following table. IVR slightly increases the running time but enhances performance.
>
> Table: The running time and MRR metric of ComplEx model on WN18RR dataset.
> |Model|Time|MRR|
> |-|-|-|
> |ComplEx|36s|0.464|
> |ComplEx-F2|39s|0.467|
> |ComplEx-N3|39s|0.491|
> |ComplEx-DURA|41s|0.484|
> |ComplEx-IVR|44s|0.494|
>
> $\textbf{Q3:}$ The captions of table 8 and 9 seem to contain an error. Should these tables be for different data sets?
>
> $\textbf{A3:}$ We will fix the errors in table 8 and table 9 in the revision. Since similar phenomena are observed in results on other datasets, we only present the results for the WN18RR dataset.

---

> ### Author Response · Authors · 2024-08-13
> **The final day for discussions**
>
> Dear reviewer:
>
> Thank you once again for your meticulous comments. As today is the final day for discussions, we kindly request your feedback once more. We anticipate that our brief responses will not require much of your time. Your feedback is invaluable to us, and we eagerly await your response.

---

> > ### Comment · Reviewer_gAVe · 2024-08-14
> > **Acknowledging your responses**
> >
> > Thank you for your response which clarifies my previous concerns. After checking reviews from other reviewers as well as corresponding rebuttal, I would like to keep my score towards acceptance.

---

### Official Review · Reviewer_fmAK · 2024-07-12

**Soundness:** 3
**Presentation:** 3
**Contribution:** 3
**Rating:** 5
**Confidence:** 4

**Summary:**

The paper addresses the challenges in Knowledge Graph Completion (KGC) using Tensor Decomposition-Based (TDB) models. The authors present a detailed overview of existing TDB models and establish a general form for these models, which is intended to serve as a foundational platform for further research and enhancement of TDB models. In addition to the new regularization technique, the paper also contributes a theoretical analysis that supports the effectiveness of their method and experimental results that demonstrate its practical utility.

**Strengths:**

1. The introduction of Intermediate Variables Regularization (IVR) is a significant original contribution. This new method addresses overfitting in tensor decomposition-based models for knowledge graph completion by focusing on the norms of intermediate variables, which is a novel angle in this field.
2. The paper also offers a theoretical analysis to support the effectiveness of IVR, presenting a comprehensive mathematical foundation that is not commonly found in many practical applications-focused papers.
3. The paper is well-organized, with clear sections dedicated to the introduction, methods, theoretical analysis, and experimental results. This structure facilitates easy understanding of the complex concepts discussed.

**Weaknesses:**

1. While the paper commendably unifies existing KGC methods through a comprehensive mathematical form, it significantly lacks an in-depth discussion on the rationale behind dividing the models into P parts, treating P more as a 'magic' parameter without a clear explanation of its theoretical or practical implications. This could leave readers questioning the basis of choosing a specific value of P and its impact on the model's performance and applicability.

2. Although the paper acknowledges the increase in parameters with the Tucker model, it does not provide a detailed theoretical or empirical analysis of these parameters. A deeper exploration into how these additional parameters affect the model’s complexity, training time, and potential for overfitting could enhance the paper's contribution and provide more actionable insights for readers looking to implement or extend the proposed methods.

3. Eq. 2 is essentially described as a special case of block-term Tucker decomposition where all Tucker's core tensors are consistent. This raises concerns regarding the originality of the design, as it may seem to be a slight variation of existing models rather than a fundamentally new approach. Expanding on how this adaptation provides unique benefits or differs in application from traditional Tucker decompositions could help in strengthening the originality aspect.

4. There is a notable absence of discussion regarding the computational efficiency of the proposed methods. Given that the introduction of IVR and the handling of increased parameters could potentially lead to higher computational costs, it would be beneficial for the paper to address these aspects. Analysis or benchmarks on the computational load compared to other models could provide a clearer picture of the practicality of implementing the proposed methods in real-world applications.

**Questions:**

1. Could you provide a more detailed explanation or theoretical basis for the division of the tensor decomposition into P parts? What are the implications of different values of P on the model's performance and how do you recommend choosing P for different scenarios?

2. Given that Formula 2 closely resembles a block-term Tucker decomposition, can you elaborate on the specific innovations or unique advantages that your adaptation provides?

3. How generalizable is the IVR method to other types of tensor decomposition models or even outside tensor-based approaches? Are there limitations in its applicability that should be considered?

**Limitations:**

The paper introduces parameters such as the number of parts (P) in the tensor decomposition but lacks a thorough analysis of how these parameters impact the overall model performance and stability. A deeper examination of the sensitivity of the model to these parameters would strengthen the discussion on limitations.

---

> ### Author Rebuttal · Authors · 2024-08-06
>
> We appreciate your careful and constructive comments. We have addressed the questions that you raised as follows. Please let us know if you have any further concerns.
>
> $\textbf{Q1:}$ Could you provide a more detailed explanation or theoretical basis for the division of the tensor decomposition into $P$ parts? What are the implications of different values of $P$ on the model's performance and how do you recommend choosing $P$?
>
> $\textbf{A1:}$ Due to space constraints, we have to put some content into the appendix. Several of your questions have been addressed there.
>
> As mentioned from Line 132 to Line 133, the number of parts $P$ determines the dimensions of the dot products of embeddings. If we treat dot product of three vectors with dimension $D/P$ as $D/P$ interactions between three vectors, then Eq.(1) results in $P^3 \times D/P = DP^2$ interactions. Thus, $P$ can be considered as a hyperparameter that controls the expressiveness of TDB models.
>
> As discussed from Line 150 to Line 154, the number of parameters in $\mathbf{W}$ equals $P^3$, and the computational complexity of Eq.(1) is $\mathcal{O}(DP^2)$. Thus, $P$ is related to expressiveness and computation. We have provided an experiment on Page 22 Paragraph “The number of Parts $P$” to study the impact of $P$ on performance. The results show that the performance generally improves and the running time generally increases as $P$ increases.
>
> Therefore, the choice of $P$ involves a trade-off between expressiveness and computation.
>
> $\textbf{Q2:}$ Provide a detailed theoretical or empirical analysis of parameters in Eq.(2) and a deeper exploration into how the additional parameters affect the model’s complexity, training time, and potential for overfitting.
>
> $\textbf{A2:}$ We have analyzed the impact of the parameter tensor $\mathbf{W}$ on model performance. As stated in Line 150 to Line 154, the number of parameters of $\mathbf{W}$ is equal to $P^3$ and the computational complexity of Eq.(1) is equal to $\mathcal{O}(DP^2)$. Table 10 on Page 21 show that the performance generally improves and the running time generally increases as the size of $\mathbf{W}$ (or $P$) increases. TuckER is a special case of Eq.(2) with $P=D$. The results show that TuckER has the best performance but the longest running time.
>
> $\textbf{Q3:}$ How Eq.(2) provides unique benefits or differs in application from traditional Tucker decompositions? Given that Eq.(2) closely resembles a block-term Tucker decomposition, can you elaborate on the specific innovations or unique advantages that your adaptation provides?
>
> $\textbf{A3:}$ The differences between traditional Tucker decomposition models and block-term Tucker decomposition Eq.(2) have been stated in Section 3.1 Paragraph “TuckER and Eq.(2)”. TuckER does not explicitly consider the number of parts $P$ and the core tensor $\mathbf{W}$, which are pertinent to the number of parameters, computational complexity and logical rules.
>
> Eq.(2) reveals the relationship between block-term TuckER decompostions and existing TDB models, which serves as a foundation for further analysis of TDB models. Although block-term Tucker decomposition has been proposed, it has not been introduced into knowledge graph completion (KGC) field. Just like exsiting TDB models only introduces tensor decomposition models into KGC field, they do not propose new tensor decomposition models. Eq.(2) presents a unified view of TDB models and helps the researchers understand the relationship between different TDB models. Moreover, the general form motivates the researchers to propose new methods and establish unified theoretical frameworks that are applicable to most TDB models.
>
> Most of your questions are about Eq.(2). We want to stress that the core contribution of our paper is the intermediate variables regularization rather than Eq.(2). Eq.(2) mainly serves as a foundation of IVR. Thus, we put more content on IVR.
>
> $\textbf{Q4:}$ Provide an analysis on the computational load compared to other models.
>
> $\textbf{A4:}$ We discuss the computational efficiency from three aspects.
>
> First, the regularization terms only increase the training time. The training time per epoch and MRR metric are shown in the following table. IVR slightly increases the running time but enhances performance.
>
> Second, we show an approach to choose the hyper-parameters in Paragraph “Hyper-parameters” on Page 20. Our approach requires only a few runs to determine optimal hyper-parameters.
>
> Third, table 10 in Page 21 show that the performance generally improves and the running time generally increases as P increases. Further analysis of other hyper-parameters is provided on Page 22 Paragraph "The hyper-parameter$\alpha$" and Paragraph "The hyper-parameter$\lambda_i$".
>
> Table: The running time and MRR metric of ComplEx model on WN18RR dataset.
> |Model|Time|MRR|
> |-|-|-|
> |ComplEx|36s|0.464|
> |ComplEx-F2|39s|0.467|
> |ComplEx-N3|39s|0.491|
> |ComplEx-DURA|41s|0.484|
> |ComplEx-IVR|44s|0.494|
>
> $\textbf{Q5:}$ How generalizable is the IVR method to other types of tensor decomposition models or even outside tensor-based approaches? Are there limitations in its applicability that should be considered?
>
> $\textbf{A5:}$ As far as we are know, all existing TDB models for KGC can be represented by Eq. (2). As stated in the Conclusion section, we intend to explore regularizations that are applicable to other types of KGC models.
>
> Our regularization, IVR, can be directly extended to other types of KGC models such as translation-based models and neural network models by incorporating the intermediate variables in these models.
>
> One limitation is the challenge of developing a theoretical framework for other types of KGC models. For TDB models, we offer a comprehensive theoretical analysis demonstrating that IVR serves as an upper bound for the overlapped trace norm. Establishing a similar theoretical foundation for other types of models remains challenging.

---

> ### Comment · Reviewer_fmAK · 2024-08-12
>
> Thank you for your response. Based on your feedback, I have decided to raise my score.

---

### Official Review · Reviewer_8dJT · 2024-07-12

**Soundness:** 3
**Presentation:** 3
**Contribution:** 3
**Rating:** 6
**Confidence:** 3

**Summary:**

The paper considers the problem of knowledge graph completion where the knowledge graph is encoded as a 3rd-order binary tensor. The authors provide an overview of existing tensor decomposition based models for KGC and propose a unifying general form that enables representing each of these models by choosing the partitioning $P$ and a core tensor $\boldsymbol{W}$ accordingly. Further, they introduce a novel regularization method called Intermediate Variables Regularization to handle overfitting in TBD models. In contrast to existing regularization approaches, the regularization term combines the norms of intermediate variables of different representations of $\boldsymbol{X}$ including its unfoldings w.r.t. each mode. Lastly, a theoretical analysis is provided followed by experiments highlighting the efficacy of IVR.

**Strengths:**

- The paper is well written and organized.
- The paper addresses an important problem and gives a broad overview of existing solutions. I think the proposed general form will facilitate future research in this research area.
- It is an intriguing approach to include and combine the norms of several intermediate variables for TDB model regularization.

**Weaknesses:**

- Currently, the paper starts directly with the main section. I think it would be beneficial to have a small (sub-)section either before or after related work to provide background on, e.g., tensors and CP/Tucker decomposition with one or two illustrating figures, and/or background on how the embedding matrices are usually obtained.

**Questions:**

- Does the choice of $P$ and $\boldsymbol{W}$ affect the performance of IVR? If yes, how?
- The paper states that IVR is applicable to most TDB models. Could you specify to which models it is not (directly) applicable?
- Are there (theoretical) cases in which IVR is expected to perform worse than existing regularization techniques?
- The justifications in the checklist are missing.

**Limitations:**

Yes.

---

> ### Author Rebuttal · Authors · 2024-08-06
>
> We appreciate your careful and constructive comments. We have addressed the questions that you raised as follows. Please let us know if you have any further concerns.
>
> $\textbf{Q1:}$ Currently, the paper starts directly with the main section. I think it would be beneficial to have a small (sub-)section either before or after related work to provide background on, e.g., tensors and CP/Tucker decomposition with one or two illustrating figures, and/or background on how the embedding matrices are usually obtained.
>
> $\textbf{A1:}$ Thank you for your valuable suggestion. We will complement the related backgroud by using the extra page in the revision. You can also refer to [1] for more related backgroud.
>
> [1] Tamara G Kolda and Brett W Bader. Tensor decompositions and applications. SIAM review, 51(3): 341 455–500, 2009.
>
> $\textbf{Q2:}$ Does the choice of $P$ and $\mathbf{W}$ affect the performance of IVR? If yes, how?
>
> $\textbf{A2:}$ Yes. We have provided an experiment in Page 22 Paragraph “The number of Parts $P$” to study the impact of $P$ on performance. The results indicate that the performance generally improves and the running time generally increases as $P$ increases.
>
> It is difficult to theoretically demonstrate how the core tensor $\mathbf{W}$ affects the performance. When $W$ is a predetermined tensor, as shown in Table 1, varying $\mathbf{W}$ leads to different performance outcomes. Establishing a clear relationship between IVR performance and $\mathbf{W}$ from Table 1 is not straightforward.
>
> When $\mathbf{W}$ is a parameter tensor, as discussed in Page 22 Paragraph “The number of Parts $P$”, the performance of IVR generally improves as the size of $\mathbf{W}$ (or $P$) increases.
>
> $\textbf{Q3:}$ The paper states that IVR is applicable to most TDB models. Could you specify to which models it is not (directly) applicable?
>
> $\textbf{A3:}$ As far as we know, IVR is applicable to all exsiting TDB models in knowledge graph completion field. However, we can not guarantee that we know all TDB models, so we claim that IVR is applicable to most TDB models.
>
> $\textbf{Q4:}$ Are there (theoretical) cases in which IVR is expected to perform worse than existing regularization techniques?
>
> $\textbf{A4:}$ Since IVR can be reduced to F2 or N3 by setting suitable hyper-parameters, IVR must perform better than F2 and N3. IVR may perform worse than DURA on some datasets or some metrics. For example, MRR metric of ComplEx-IVR is slightly worse than that of ComplEx-DURA on FB15k-237 dataset.
>
> $\textbf{Q5:}$ The justifications in the checklist are missing.
>
> $\textbf{A5:}$ We will add the justifications in the revision.

---

> > ### Comment · Reviewer_8dJT · 2024-08-12
> >
> > I thank the authors for the response. After reading the rebuttal and the other reviews, I will keep my score.

---

### Author Response · Authors · 2024-08-11
**Responses for Reviewers**

Dear reviewers:

Thanks again for your constructive comments. We appreciate your time and effort in reviewing our paper. Since the final day of discussion is coming, we sincerely look forward to your further feedback. Due to our brief responses, it will not take up too much of your time. If you find that our responses have effectively addressed your concerns, can you give us feedback? Your feedback is crucial to us. We look forward to hearing from you.

---

### Author Response · Authors · 2024-08-14
**Thanks for the feedback**

Dear reviewers:

We sincerely appreciate your insightful comments and valuable feedback.

---

### Decision · Program_Chairs · 2024-09-25

**Decision:**

Accept (poster)

**Comment:**

The paper presents an interesting perspective on enhancing Knowledge Graph Completion (KGC) with Tensor Decomposition-Based (TDB) models. During the rebuttal phase, the authors have addressed most of the concerns. After discussing with the SAC, we come to an agreement that the idea of regularizing intermediate variables is novel, and the performance improvements are significant. The community might get inspired by this paper to carry out future research in this direction.

Consequently, I recommend accepting this paper as an poster. Nonetheless, the authors are required to polish the main paper according to their response in the rebuttal phase.